# A unifying mechanism for cation effect modulating C1 and C2 productions from $CO_2$ electroreduction

Seung-Jae Shin [1,7], Hansol Choi [2,7], Stefan Ringe [3], Da Hye Won [4], Hyung-Suk Oh [4], Dong Hyun Kim[5], Taemin Lee[6], Dae-Hyun Nam [6], Hyungjun Kim [1] ✉ & Chang Hyuck Choi [5] ✉

Electrocatalysis, whose reaction venue locates at the catalyst–electrolyte interface, is controlled by the electron transfer across the electric double layer, envisaging a mechanistic link between the electron transfer rate and the electric double layer structure. A fine example is in the $CO_2$ reduction reaction, of which rate shows a strong dependence on the alkali metal cation ($M^+$) identity, but there is yet to be a unified molecular picture for that. Using quantum-mechanics-based atom-scale simulation, we herein scrutinize the $M^+$-coupling capability to possible intermediates, and establish $H^+$- and $M^+$-associated ET mechanisms for $CH_4$ and $CO/C_2H_4$ formations, respectively. These theoretical scenarios are successfully underpinned by Nernstian shifts of polarization curves with the $H^+$ or $M^+$ concentrations and the first-order kinetics of $CO/C_2H_4$ formation on the electrode surface charge density. Our finding further rationalizes the merit of using Nafion-coated electrode for enhanced C2 production in terms of enhanced surface charge density.

Ever-increasing global energy demand is bringing imbalances in the natural cycles, and critically threatening our sustainability. Towards a technological breakthrough that can restore sustainable cycles by rectifying those imbalances, electrochemical $CO_2$ reduction reaction ($CO_2$RR) is considered promising for converting $CO_2$ into valuable chemicals[1–3], such as CO, $CH_4$, and $C_2H_4$. Coinage-metal electrodes of Ag, Au, and Cu are known to be active towards the $CO_2$RR[4–7], on which intense research efforts have been focused[8–13], while a full understanding of the $CO_2$RR mechanism on these catalysts is still not yet at hand.

The most intriguing question is the mechanistic role of alkali metal cations ($M^+$) at the catalyst–electrolyte interface. Opposed to the conventional view that $M^+$ would spectate the reaction, many experimental reports have evidenced a noticeable dependence of $CO_2$RR

activity and selectivity on the $M^+$ identity[14–17], often termed a cation effect. Specifically, the selectivity trend towards the C2 product and the CO production activity trend were found to follow the order of $Cs^+ > Rb^+ > K^+ > Na^+ > Li^+$[14,15], which clearly indicate a certain mechanistic role of $M^+$ at the rate-determining step (RDS).

Supposing the proton-coupled electron transfer (PCET) as the RDS, the origin of the cation effect was ascribed to the variation in the local pH at the interface due to the different pKa values of the buffering $M^+$[15]. However, various other experiments, where no clear activity change was observed upon (bulk) pH variations, demonstrated the possibility that the RDS involves no proton transfer[12,18–21]. Meanwhile, Chan group suggested a mechanism for the cation effect, called a field effect[16,17,22,23]: the electric double layer (EDL) field formed across the Helmholtz layer would stabilize the intermediates (e.g., $^*CO_2$) via

[1]Department of Chemistry, Korea Advanced Institute of Science and Technology, Daejeon 34141, Republic of Korea. [2]School of Materials Science and Engineering, Gwangju Institute of Science and Technology, Gwangju 61005, Republic of Korea. [3]Department of Chemistry, Korea University, Seoul 02841, Republic of Korea. [4]Clean Energy Research Center, Korea Institute of Science and Technology, Seoul 02792, Republic of Korea. [5]Department of Chemistry, Pohang University of Science and Technology (POSTECH), Pohang 37673, Republic of Korea. [6]Department of Energy Science and Engineering, Daegu Gyeongbuk Institute of Science and Technology, Daegu 42988, Republic of Korea. [7]These authors contributed equally: Seung-Jae Shin, Hansol Choi. ✉e-mail: linus16@kaist.ac.kr; chchoi@postech.ac.kr

adsorbate dipole-field interaction, which can be modulated by the degree of $M^+$ accumulation at the interface[17,24]. Also, using the scanning electrochemical microscopic technique, the Koper group demonstrated an absence of $CO_2RR$ activity for CO formation without $M^+$, which led them to propose a mechanism based on a $M^+$-complexation (or coupling) to the $CO_2^-$ intermediate in conjunction with their ab initio molecular dynamics (AIMD) simulation results[13]. Moreover, cation-dependent interfacial water structure has been exploited to understand the cation effect on the $CO_2RR$, which yields different electric field strengths[25,26], adsorption rate[27], or surface-dependent solvation structure[28]. There further exist other general discussions on the cation effect to the electrocatalytic activity[29], $e.g.$, site-blocking of reactants on the electrode or surface reconstruction, albeit it has not been directly linked with the cation effect on the $CO_2RR$.

Towards the definition of a general scheme, at the molecular level, it is thus utmostly required to establish a unified $CO_2RR$ mechanism based on a systematic assessment, which can elucidate the cation effect on the activity for the CO formation and selectivity towards C–C coupling for multicarbon products. Herein, we investigate the cation-controlled mechanism by reflecting more practical electrolysis conditions via full-equilibrium simulations and flow-type-electrolyzer experiments with gas diffusion electrodes (GDE). Using a quantum-mechanics-based multiscale simulation, offering an accurate description on the EDL structure at atom-scale[30], we mapped out the cation-coordinating ability to all possible intermediates and established corresponding reaction energy profiles for CO, $CH_4$, and $C_2H_4$ productions. The suggested different nature of RDSs, either a cation-coupled electron transfer (CCET) step for CO and $C_2H_4$ productions or a PCET for $CH_4$ production, was corroborated by our experiments widely varying the $M^+$ concentrations and identities. Of prime interest is that the cation effect results from a cation-dependent electrode surface charge density ($|\sigma|$). Our mechanism further successfully accounts for recent empirical but breakthrough findings ($e.g.$, ionomer effects)[6,31–35], and highlights the importance of EDL engineering as the next quest for better $CO_2$ electrolysis.

## Results and discussion

### To be, or not to be coordinated by a cation
To identify the chemical role of $M^+$ during $CO_2RR$, we first investigate atomic details of the catalyst–electrolyte interface using density functional theory in classical explicit solvent (DFT-CES) simulation[36]. This method offers an accurate description of the electrified interface at a balanced computational cost, by mean-field coupling of a quantum mechanical description on the catalyst surface with a molecular dynamics description on the liquid structure of the electrolyte phase[37]. Recent advances in computational simulations enable a direct investigation of the electrode-electrolyte structure, highlighting the importance of the atomic arrangement of EDL constituents (e.g. chemisorbed water, Helmholtz ions, etc.) at the buried nanoscale region, which can affect the catalytic reactions under the actual electrochemical conditions[30,38–44]. Compared with the AIMD simulation, often used for modeling the electrochemical interfaces, the DFT-CES enables to investigate electrolyte phase dynamics with many more atoms over a more extended time-scale; multi-thousands of atoms over a few nanoseconds using the DFT-CES $vs.$ multi-hundreds of atoms over a few picoseconds using the AIMD[13,38,40,41,45]. The availability of simulations at full length- and time-scales is critical to unbiasedly confirm the possible coordinating ability of electrolyte constituents to the intermediate species since it can provide the full equilibrium-dynamic structure of the electrolyte phase without the influence of initial conditions. Most importantly, we note that the DFT-CES succeeded in unraveling the atomic origin of the famous camel-shaped behavior of the EDL capacitance, confirming its accuracy in describing the EDL structural details[30].

Using DFT-CES simulations, we investigate the cation-coordinating ability of 28 possible intermediate species that can be formed during the reaction paths of $CO_2RR$ (Fig. 1 and Supplementary Fig. 1); a path to form CO (Fig. 1b; blue), $CH_4$ (Fig. 1c, green), and $C_2H_4$ via a C–C coupling step (Fig. 1d, red). Cu(100) surface, known to be active for C–C coupling reactions[46], was chosen as the model catalyst surface for $CH_4$ and $C_2H_4$ formation paths, as well as Ag(111) surface for CO formation path. At two different potentials of $-0.5$ V vs. standard hydrogen electrode (SHE) for the potential at point of zero charge ($E_{PZC}$) and $-1.0$ $V_{SHE}$ for the interface charge of $-18$ $\mu C$ $cm^{-2}$ (Supplementary Fig. 2), DFT-CES simulations identified 6 intermediate species $-^*CO_2$, $^*COOH$, $^*CHO$, $^*OCCO$, $^*OCCOH$, and $^*HOCCOH$–to be coordinated by a cation; herein $K^+$ (Fig. 1a and Supplementary Figs. 3–5).

After constructing reaction paths with explicitly specifying the cation-coordinated intermediate species (as illustrated in Fig. 1b–d), we calculated the reaction free energy profile of each reaction path as shown in Fig. 2a–c (Supplementary Figs. 6 and 7 for all intermediates; see Supplementary Note 1 for the computational details). Full reaction free energy profiles suggest the kinetics of CO, $CH_4$, and $C_2H_4$ formations, to be respectively controlled by the RDSs of

$$^* + CO_2 + M^+ + e^- \rightarrow {}^*COO^- \cdots M^+ \tag{1}$$

$$^*CO + H^+ + e^- \rightarrow {}^*COH \tag{2}$$

$$^*2CO + M^+ + e^- \rightarrow {}^*OCCO^- \cdots M^+ \tag{3}$$

Here, reaction (2) is usually termed a PCET, and thus in an analogical sense, reactions (1) and (3) can be termed a CCET.

The proposed RDSs corroborate previous experiments. Previous studies demonstrated a strong dependence of CO formation or C–C coupling rates on the cation identity, $i.e.$, cation effect[14–16,47,48]. Also, Monteiro et al. showed a lack of $CO_2RR$ activity to CO without $M^+$, which initiated an intensive discussion about the possibility of CCET[13]. In addition, Chan and coworkers investigated the kinetic importance of proton activity using pH control experiments[12,46]. They found that the pH variation significantly changes the $CH_4$ production rate[46], while the CO and $C_2H_4$ production rates are nearly unchanged on a SHE potential scale[12,46]. The reactions (1) and (3) infer a critical role of $M^+$ in the kinetics of CO and $C_2H_4$ formation paths, and reaction (2) shows the kinetic importance of pH in the $CH_4$ formation path.

### Nature of cation-coupled electron transfer
Although the CCET is named after the PCET, there is a caveat to understand the nature of CCET in parallel to the PCET. Since cations other than a proton are much heavier than an electron, the non-adiabatic effect can no longer play a role in determining the transfer rate[49]. Rather, it is more reasonable to consider a Born-Oppenheimer-type picture, where an electron is adiabatically transferred to the intermediate species along the reaction coordinate for the cation-coupling.

Analysis of the electronic response of the catalyst–adsorbate system during DFT-CES iterations provides valuable insight on the nature of CCET, which is indeed an adiabatic response of the electron density upon the electrolyte structure change. Figure 2d shows the change of cation coordination number (CN) to the key intermediate species of $^*CO_2$ and $^*OCCO$, and the change of their partial charges. We find no cation-coupling at the $0^{th}$ iteration step, but the electron density between the metal and the adsorbate is redistributed during iterations, which enables the cation-coupling (Fig. 2e and Supplementary Fig. 8). Projected density of states (PDOS) shows that the lowest unoccupied molecular orbital (LUMO) of the adsorbate is downshifted after the cation-coupling due to the field generated by the cation (Supplementary Fig. 9). This increases the electronic

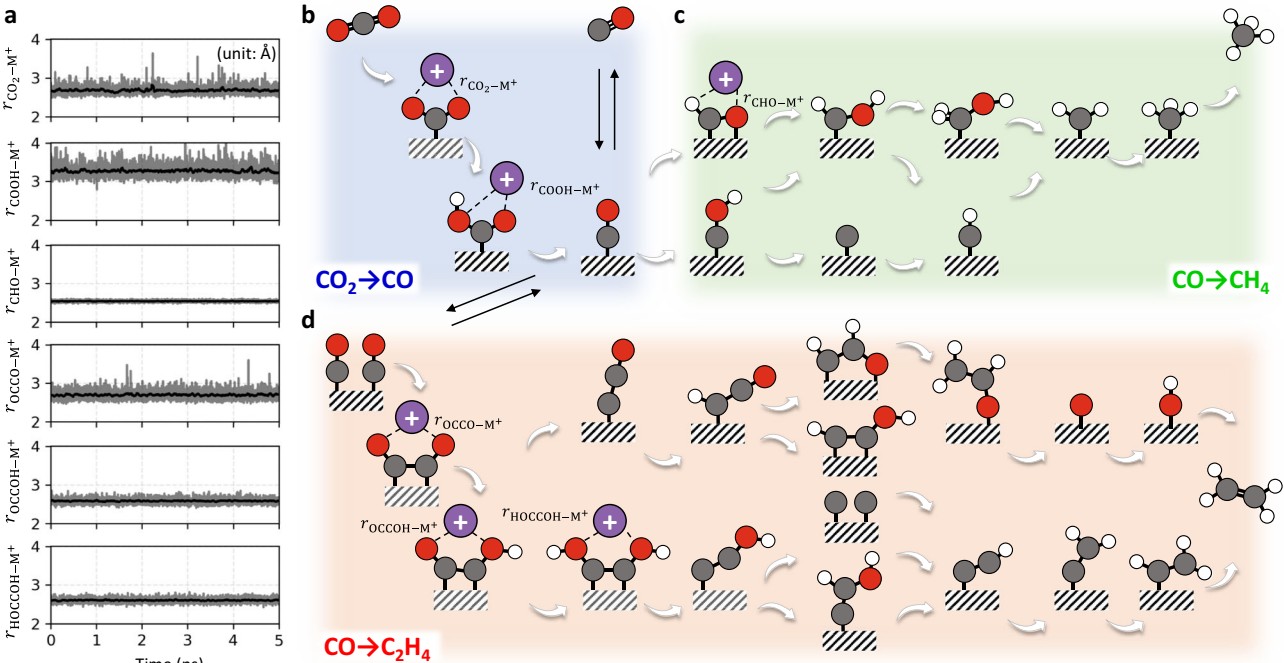

**Fig. 1 | Map of cation-(un)coordinated intermediates during CO₂RR. a** Temporal change of distance (*r*) between intermediate and K⁺ during 5-ns DFT-CES trajectory (grey line). Applied potential is −1.0 V$_{SHE}$. The black solid line denotes a moving average of the grey line using a 0.05-ns time window. **b–d** Possible intermediates for CO₂RR, where cation-coordinated species are identified with omitting the water molecules for visual clarity. Reaction paths are denoted using different background colors; blue for CO₂-to-CO (**b**), green for CO-to-CH₄ (**c**), and red for CO-to-C₂H₄ (**d**). Cation CN to each intermediate is provided in Supplementary Fig. 4, and the species with CN ‑ 1 are identified as cation-coordinated intermediates. All cartooned structures are based on the DFT-CES simulation (Supplementary Fig. 1). The dark-grey, red, white, and purple colored ball indicates the C, O, H, and K⁺, respectively, and the electrode is shown with the hatch patterned box.

occupation of LUMO[50], which partially reduces the adsorbate species (Fig. 2f), yielding the partial charges of *CO₂ and *OCCO to be −0.7 and −0.9, respectively. The results of DFT-CES simulation are in agreement with the previous novel findings showing that the alkali metal cations can stabilize the *CO₂ intermediate[13,51] and *OCCO intermediate[52,53], respectively.

The entire electron density redistribution, which is associated with the adiabatic reaction coordinate not only for the cation-coupling but for the adsorbate-binding[50], can be conceptualized in two different pictures; either an electronic polarization or an electron transfer from metal to adsorbate. For the *CO₂ case, the former concept implies an absence of ET at the RDS;

$$* + CO_2 + M^+ \rightarrow {}^*CO_2 \cdots M^+ \qquad (4)$$

which is followed by a subsequent fast ET[50]. Electrostatically, the polarization induces a dipole that can be stabilized by an external field. Thus, the field effect perspective, suggested by Chan group[16,17,22,23], can be further elaborated by identifying the atomic structural details of the cation that generates the field to stabilize the dipole induced at the metal-intermediate interface.

On the other hand, the latter concept of electron transfer literally implies that the intermediate such as *CO₂ should be reduced into *CO₂⁻ at the final stage of the adiabatic reaction path of adsorption and cation-coupling, i.e., the reaction (1) becomes an appropriate expression for the RDS. Although this is similar to what is suggested by Koper group[13], they illustrated a stepwise path of reductive adsorption and cation-coupling. The ET perspective can be further supported by the intermediate partial charge close to −1, and the PDOS demonstrating an electron-filling to the downshifted LUMO after the cation-coupling. However, strictly speaking, electrons exist as a cloud of indistinguishable quantum-mechanical particles. Consequently, the distinction between polarization and transfer depends on a hypothetical

partitioning of the electron density in space, and both are the same phenomenon if there is a significant electronic overlap between the metal and the adsorbate[54]. Thus, there is no fundamental difference between the reaction (4) + fast ET and the reaction (1), but they are two different viewpoints on the same phenomenon; the former stems from more continuum-level and electrostatic perspective, while the latter stems from more atomic-level and charge-transfer perspective.

### Cation concentration-dependent Nernstian shifts

To elucidate the proposed CCET mechanism, we investigated CO₂-to-CO conversion on the polycrystalline Ag electrode in various concentrations of KOH electrolytes (0.01–10 M) using a flow cell reactor (see Methods, Supplementary Figs. 10 and 11). The CO₂RR polarization results are provided in Fig. 3a–c (with respect to different reference potential scales; Supplementary Fig. 12 for the Faradaic efficiency (FE)). On both SHE (Fig. 3a) and reversible hydrogen electrode (RHE, Fig. 3b) scales, a partial current density of CO ($j_{CO}$) shows considerable deviations in their polarization curves and is promoted as the KOH concentration increases. The departure of $j_{CO}$ curves in SHE and RHE scales implies that CO₂RR kinetics does not simply depend on the electrode potential (i.e., * + CO₂ + e⁻ → *CO₂⁻), nor does its RDS accompanies the PCET step (i.e., * + CO₂ + H⁺ + e⁻ → *COOH), reasonably leading us to account for K⁺-coupled mechanism in their RDSs on the basis of our simulation results.

Hence, we re-plotted the polarization curves with respect to an alkali metal cation concentration-corrected electrode (CCE) scale (Fig. 3c), defined here as $E_{CCE} = E_{SHE} - 0.059 \times \log[M^+]$, where [M⁺] denotes a bulk cation concentration[55]. This plot identifies a collapse of the $j_{CO}$ polarization curves independent of the KOH electrolyte concentration, corresponding to a Nernstian potential shift of *ca.* 60 mV per log[K⁺] on the SHE scale. An identical trend was also confirmed in 0.01 M KOH + 0–0.495 M K₂CO₃ electrolytes (Supplementary Figs. 13 and 14), in which only the K⁺ concentration was

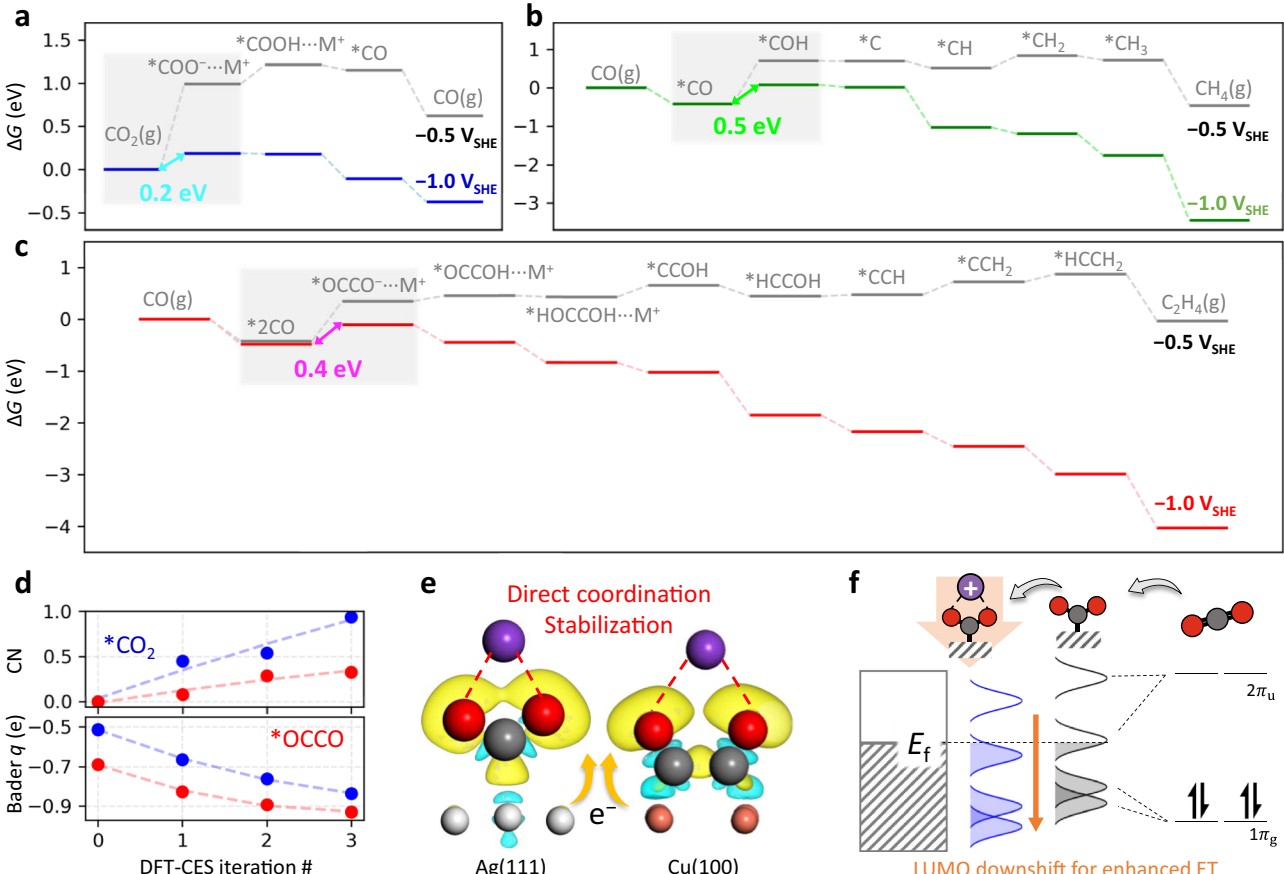

**Fig. 2 | Reaction energy diagrams and cation-coupled electron transfer. a–c,** Reaction free energy ($\Delta G$) diagrams for $CO_2$-to-CO on Ag(111) (**a**), CO-to-$CH_4$ on Cu(100) (**b**), and CO-to-$C_2H_4$ on Cu(100) (**c**). Applied potentials are −0.5 and −1 $V_{SHE}$. The asterisk (*) indicates the adsorbed state, and the energetically favorable paths, among other possibilities (Supplementary Fig. 6) are selectively shown. The cation-coordinated intermediates are specified by appending ···$M^+$. The RDS is denoted by a grey-shaded region. **d,** Change in CN of $K^+$ to the *$CO_2$ and *OCCO, and that in the Bader charge (Bader $q$) of those two intermediates during DFT-CES iterations at −0.5 $V_{SHE}$. **e,** Cation coupling to the intermediates not only stabilizes them, but also helps an electron transfer to them from the metal electrodes. Iso-surface shows a charge difference between the cation-uncoupled case at the 0th iteration step and the cation-coupled case at the last iteration step (isosurface value is 0.0006 e $a_0^{-3}$). Yellow and cyan colors indicate charge accumulation and depletion, respectively. The water molecules are omitted for visual clarity. **f,** A schematic showing the energy level change of $CO_2$, which is based on the PDOS analyses (Supplementary Fig. 9). Before the adsorption, $CO_2$ at a linear form has $1\pi_g$ HOMO state and $2\pi_u$ LUMO state. When it is adsorbed to the metal, the states are broadened due to the mixing with metal states, and the bent form breaks the degeneracy of $2\pi_u$ and $1\pi_g$ states; one level shifts down from the $2\pi_u$ LUMO state and is partially filled, yielding a radical character. When a cation is coupled to the bent *$CO_2$, the LUMO state downshifts further due to the electric field from the cation. This increases the electronic occupation to the *$CO_2$, yielding an anionic character of *$CO_2$. $E_f$ denotes the Fermi level.

varied but electrolyte pH was almost untouched (Supplementary Fig. 15). With a Tafel slope of 120–130 mV dec$^{-1}$ for the $j_{CO}$, the results support that the RDS for CO formation path is the first ET step involving the $K^+$-coupling, i.e., the reaction (1). Notably, the collapse of $j_{CO}$ curves in the CCE scale is not a singular event that occurs limitedly on Ag electrode in the alkaline electrolyte, but can also be found in other representative electrodes for efficient CO production, e.g., Au and NiNC (Supplementary Figs. 16–19). However, the $CO_2$RR in acidic electrolytes is unable to be explained together due to a partial displacement of cations by the protons (or hydronium ions)[56] (Supplementary Fig. 20).

Afterwards, the $C_2H_4$ formation path, which was also predicted to follow the CCET step, was investigated. Herein, instead of the $CO_2$RR, CO reduction reaction (CORR) was chosen as a model reaction for clearer reaction kinetic studies on a polycrystalline Cu electrode (Supplementary Fig. 21; see Supplementary Note 2 and Supplementary Figs. 22–25 for $CO_2$RR on the Cu electrode). CORR was also performed in various electrolytes having different pHs and K$^+$ concentrations, i.e., 0.5–5 M KOH (Fig. 3d–i) and 0.5 M KOH with 0.25/2.25 M $K_2CO_3$ (Supplementary Figs. 26 and 27), and the partial

current density of ethylene ($j_{C2H4}$) was plotted with respect to the SHE, RHE, and CCE scales. As the $j_{CO}$ trend, the results exhibited a collapse of the $j_{C2H4}$ curves on the CCE scale, but marked departures on the SHE and RHE scales, with a Tafel slope of *ca.* 120 mV dec$^{-1}$ (Fig. 3d–f). Therefore, RDS for the $C_2H_4$ formation of CORR is also identified as the first ET step coupled with one $K^+$ transfer, i.e., the reaction (3).

On the other hand, the partial current densities of $CH_4$ ($j_{CH4}$), measured by CORR on the Cu electrode, are collapsed on the RHE scale, but considerable deviations can be seen on the SHE and CCE scales (Fig. 3g–i). Their Tafel slopes are *ca.* 120 mV dec$^{-1}$, indicating that RDS of the $CH_4$ formation from CORR is the first ET step via PCET, i.e., the reaction (2). Therefore, our experimental findings for all CO, $C_2H_4$, and $CH_4$ formation paths greatly support the DFT-CES predictions that the two formers accompany the CCET while the latter does the PCET in their RDSs.

## CCET kinetics controlled by surface charge density
Besides the electrolyte pHs and $K^+$ concentrations, $CO_2$RR activity or selectivity is known to be affected by the $M^+$ identity, i.e., a cation

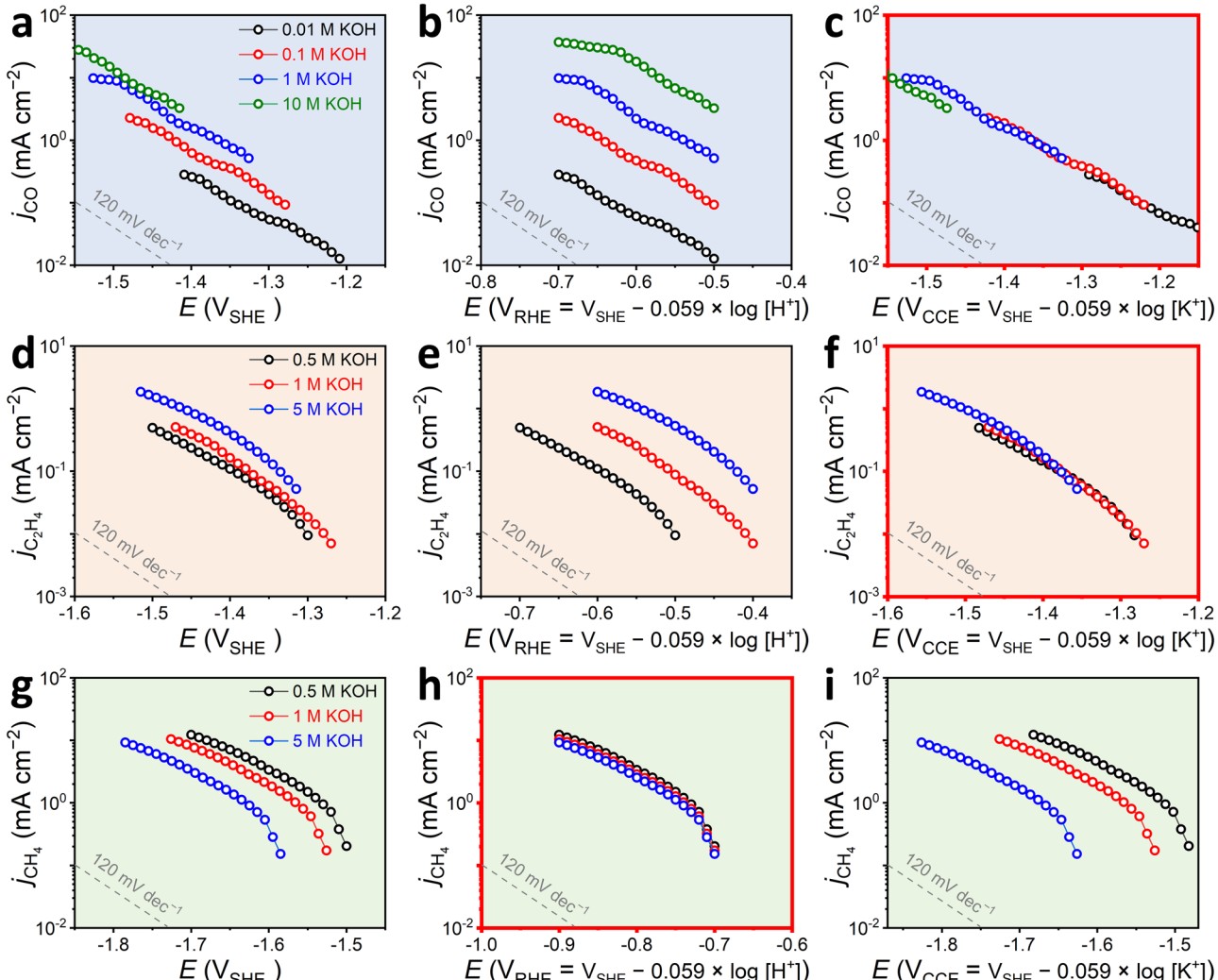

**Fig. 3 | Electrochemical CO₂RR and CORR in various electrolytes. a–c** The CO₂-to-CO conversion was measured on the Ag electrode in 0.01–10 M KOH electrolytes. **d–f** The CO-to-C₂H₄ and **g–i** the CO-to-CH₄ conversions were measured on the Cu electrode in 0.05–5 M KOH electrolytes. All the electrolysis measurements were performed in an electrochemical flow cell. The polarization curves are plotted with respect to the **a, d, g** SHE, **b, e, h** RHE ($V_{SHE} - 0.059 \times log[H^+]$), or **c, f, i** CCE ($V_{SHE} - 0.059 \times log[K^+]$) scales. The grey lines indicate a Tafel slope (typically plotted as an inverse function of the present polarization curve) of 120 mV dec⁻¹. Collapses of the polarization curves are found for the CO and C₂H₄ formations upon the CCE scale and for the CH₄ formation upon RHE scale, which are highlighted with red boxes.

effect[14,15]. The M⁺-dependent CO₂RR activity was also reproduced in our experiments, performed on Ag (Supplementary Fig. 28) and Cu (Supplementary Fig. 24) electrodes. They show activity trend of Cs⁺ > Rb⁺ > K⁺ > Na⁺ > Li⁺ for CO and C₂H₄ formations but opposite trend for CH₄ formation. An identical trend was also found for CORR on Cu electrode (Supplementary Fig. 29).

Cation-dependent activity change could be ascribed to the different intermediate stabilization ability of different M⁺ at the RDS, where the cation is coupled. However, the larger cation has a longer coordination distance, when it develops an inner-sphere interaction with the negatively charged intermediate (*e.g.*, *COO⁻ or *OCCO⁻), resulting in a less coulombic stabilization of the intermediate[13], and thus predicting an activity trend opposite to that of the experiment.

Instead, reaction kinetic study, which can provide definite evidence on reaction mechanism[57], unravels that different CO or C₂H₄ production rates depending on the M⁺ identity (and its bulk concentration) are primarily attributed to different |σ|. Considering that the CCET steps of the reactions (1) and (3) govern overall CO and C₂H₄ production rates, respectively, the Butler-Volmer kinetics

at large cathodic overpotentials yield

$$j_{CO} = n_1 F k_1 P_{CO_2} C_{M^+} e^{-\alpha_c F \left(E - E^{0'}_{(Reaction\,(1))}\right)/RT} \quad (5)$$

$$j_{C2H4} = n_2 F k_2 P_{CO}^2 C_{M^+} e^{-\alpha_c F \left(E - E^{0'}_{(Reaction\,(3))}\right)/RT} \quad (6)$$

where $F$, $R$, and $T$ are the Faraday constant, gas constant, and temperature, respectively. $n_{1(2)}$ and $k_{1(2)}$ are the number of electrons and rate constant involved in the CO₂-to-CO (or CO-to-C₂H₄) conversion reaction, respectively, and $P_{CO_2(CO)}$ denotes the CO₂ (CO) partial pressure. $E^{0'}_{(Reaction(1))}$ and $E^{0'}_{(Reaction(3))}$ are the formal reduction potential of the elementary step reaction (1) and reaction (3), respectively, and $\alpha_c$ is the cathodic charge transfer coefficient.

According to the equations, the reaction rates are determined by the local cation concentration at the interface, $C_{M^+}$. Unfortunately, this parameter is not straightforwardly measurable or even defined[58,59]. Instead, it can be reasonably hypothesized that the excess cations at the EDL[59], which locates there to screen the electrode surface charge,

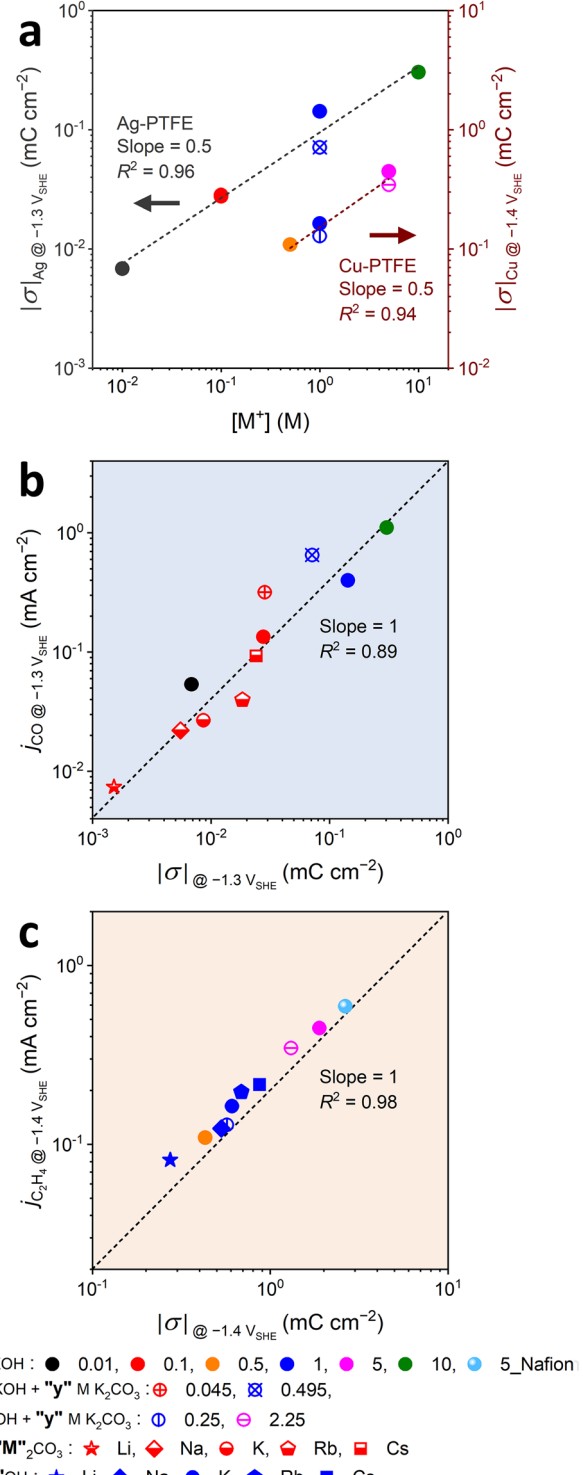

**Fig. 4 | Reaction kinetic studies. a** Correlation plots between $[M^+]$ and $|\sigma|$ for the Ag and Cu electrodes. Correlation plots of **b** $j_{CO}$ on the Ag electrode and **c.** $j_{C2H4}$ on the Cu electrode with $|\sigma|$. The data was collected at −1.3 $V_{SHE}$ for the Ag electrode ($CO_2$-to-CO formation) and −1.4 $V_{SHE}$ for the Cu electrode (CO-to-$C_2H_4$ formation). Electrolytes used for gathering these data can be classified into five different categories: KOH electrolytes with different concentrations (filled circles), 0.01 M KOH electrolytes with additional $K_2CO_3$ salt (double-crossed circles), 0.5 M KOH electrolytes with additional $K_2CO_3$ salt (single-crossed circles), 0.05 M $M_2CO_3$ electrolytes (half-filled symbols), and 1 M MOH electrolytes (filled symbols), where M is Li, Na, K, Rb, and Cs. In addition, a result collected on the Nafion-coated Cu electrode in 5 M KOH electrolyte (a filled circle with pale blue color) was also provided. Here, the total $M^+$ concentration and its identity are distinguished by color and symbol shape, respectively. Guidelines for the slopes are indicated by a dashed line.

The staircase potentio electrochemical impedance spectroscopy (SPEIS) reveals magnified $C_{diff}$ values as $[M^+]$ increases or cation size becomes larger (Supplementary Figs. 31 and 32), consequently leading to a wide $|\sigma|$ range, 0.001–0.304 mC cm$^{-2}$ at −1.3 $V_{SHE}$ for Ag electrode and 0.082–0.447 mC cm$^{-2}$ at −1.4 $V_{SHE}$ for Cu electrode. A correlation between $|\sigma|$ and $[M^+]$ identifies that $|\sigma|$ is proportional to $[M^+]^{0.5}$ (Fig. 4a), which agrees with the simple prediction using the Gouy-Chapman theory[61]. Notably, their relationship greatly rationalizes the collapse of kinetically described $j_{CO}$ or $j_{C2H4}$ upon thermodynamically (or Nernstianly) $M^+$ concentration-corrected potential (*i.e.*, CCE) scale as shown in Fig. 3 (see Supplementary Note 3 for detailed discussion), inferring their RDS to be involved with the CCET path.

More evidently, both a correlation plot between $j_{CO}$ and $|\sigma|$ at −1.3 $V_{SHE}$ and that between $j_{C2H4}$ and $|\sigma|$ at −1.4 $V_{SHE}$ show a slope of unity in the logarithmic scale (Fig. 4b, c). The first-order kinetics of $j_{CO}$ and $j_{C2H4}$ on $|\sigma|$ clearly demonstrates our mechanism again that their RDSs accompany the CCET step, i.e., reactions (1) and (3). Considering that data gathered with various $M^+$ identities locates on the linear correlation curve of $j_{CO/C2H4}$ and $|\sigma|$, which was plotted from all other control experiments, the changes in $|\sigma|$ mostly ascribes the changes in both $j_{CO}$ and $j_{C2H4}$. This enables us to conclude that the cation effect on CO and $C_2H_4$ formations to be controlled by the $|\sigma|$. However, the atomic origin of its cation-species dependence needs to be further unraveled; cation-specific interfacial solvation could play a significant role[13,25–29,51], suggesting a future research direction.

**Tuning the surface charge density for enhanced C−C coupling**
On the basis of above understandings, it can now be rationalized why great $C_2H_4$ productions have been exclusively reported so far with highly concentrated MOH electrolytes (>1 M)[6,31,34]. At the same potential on the SHE scale, high pH electrolyte is not only beneficial for suppressing proton activity and consequent PCET pathways (*e.g.*, methane and $H_2$ formation), but also induces high $|\sigma|$, which is indispensable for stabilizing *OCCO$^-$ intermediate and thereby lowering energy cost for the C−C coupling step.

More interestingly, we can further provide a clue to a fundamental origin of modulated $CO_2RR$ activity and selectivity in the presence of ionomer at the interface, highlighted very recently with boosted $C_2H_4$ and other C2 product formations on the Nafion-coated electrode[6,31–35]. In literature, these empirical findings have been deemed as a result from either high local pH (induced by accumulation of OH$^-$ ions at the electrode–ionomer interface) and consequent high $CO_2$ concentration or better mass transport of ionic species. Similarly, we also found 1.6 times higher $j_{C2H4}$ (0.59 mA cm$^{-2}$ at −1.4 $V_{SHE}$) on Nafion-coated Cu electrode during CORR (and $CO_2RR$, Supplementary Fig. 33) in a 5 M KOH electrolyte than that on bare Cu electrode (Fig. 5a and Supplementary Fig. 34). Also, as shown in Fig. 5b, their SPEIS results verified a significantly tuned $C_{diff}$ value (−2 mF cm$^{-2}$) on the Nafion-coated electrode, which was *ca.* 1.6 times larger than that on the bare electrode (−1.2 mF cm$^{-2}$). Surprisingly, the $j_{C2H4}$ and estimated $|\sigma|$ values for the

will involve in the CCET reaction. If this assumption is true, the CO and $C_2H_4$ production rates should show the first-order reaction kinetics on the $|\sigma|$ at the same $P_{CO_2(CO)}$ and $E$ on the SHE scale, because $C_{M^+}$ will be equal to or (at least) proportional to the $|\sigma|$[59].

The $|\sigma|$ at a certain potential ($E'$) can be estimated by integrating differential capacitance ($C_{diff}$) from the $E_{PZC}$ (Supplementary Fig. 30), using the following equation[60].

$$|\sigma| = |\int_{E_{PZC}}^{E'} C_{diff} \, dE| \qquad (7)$$

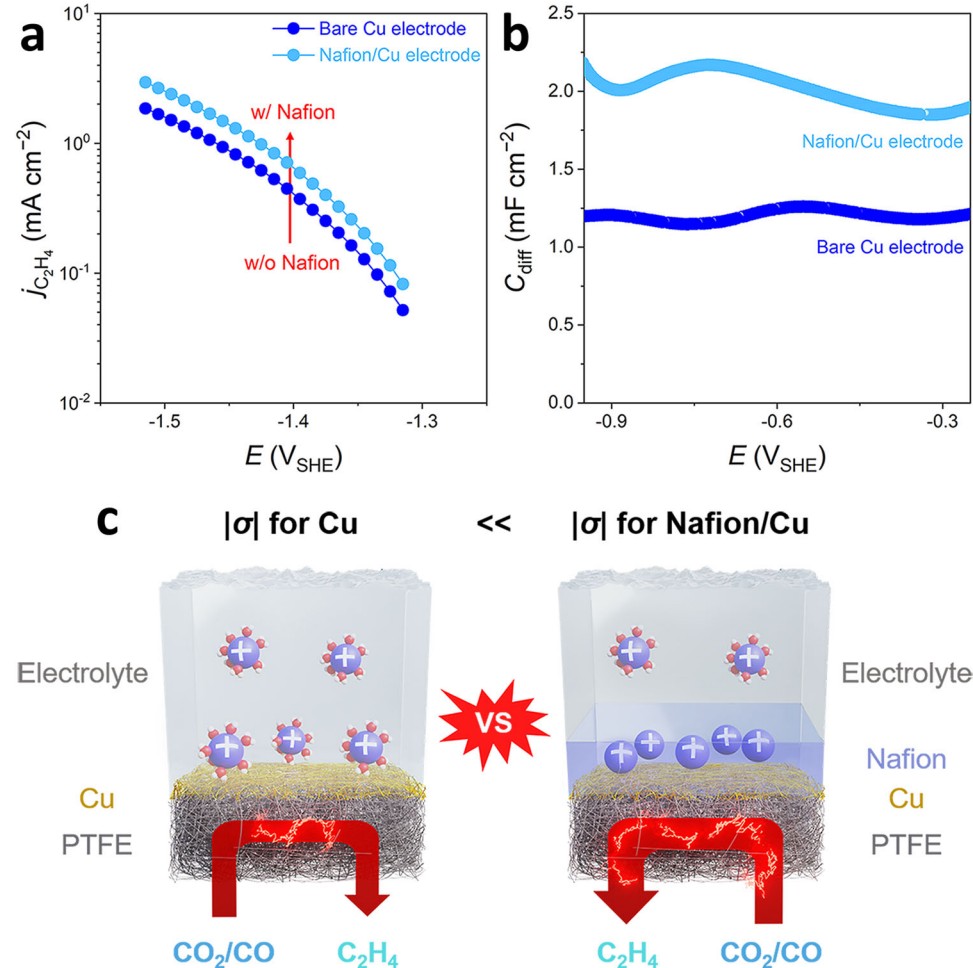

**Fig. 5 | Boosted C₂H₄ formation on the Nafion-coated Cu electrode. a** The $j_{C2H4}$ *vs.* potential curves and **b** the $C_{diff}$ curves, measured on bare and Nafion-coated Cu electrodes in 5 M KOH electrolyte. **c** Schematic descriptions of the catalyst–electrolyte interface for the Nafion-free and Nafion-coated Cu electrodes, showing a higher $|\sigma|$ on the Nafion-coated one.

Nafion-coated electrode lie exactly on a previous trend line in Fig. 4c made upon the bare Cu electrode, unveiling that fundamental origin of the boosted C₂H₄ production with Nafion ionomer is due to a higher $|\sigma|$ and consequent promotion of the reaction (3) via the CCET pathway (Fig. 5c).

In summary, we present a CCET-based mechanism for CO₂RR that identifies the role of cations in modulating the activity and selectivity during CO₂RR towards CO, CH₄, and C₂H₄ formations. Atomic- and electronic-level elucidation on the catalyst–electrolyte interfacial region, empowered by DFT-CES, helps our understanding on the nature of CCET, corroborating previous experimental findings, and mechanistic suggestions. In addition, we demonstrate distinct Nernstian shifts depending on the bulk cation concentration, and first-order kinetics on the electrode surface charge density, both of which evidence our CCET-based mechanism. Most interestingly, our kinetic study finds the cation effect results from the cation-dependent electrode surface charge density. This understanding not only accommodates past and present efforts to tune the electrochemical interfaces for improved CO₂RR electrolysis (e.g., high pH operation and ionomer-coating) but also brings up a strategic discussion to maximize the electrode surface charge density for further improvements.

## Methods
### DFT-CES simulations
DFT-CES is a grid-based mean-field theory for the quantum mechanics/molecular mechanics (QM/MM) multiscale simulations[36], where the interfacial interaction is developed based on QM energetics. This is implemented in our in-house code that combines the Quantum ESPRESSO density functional theory simulation engine and LAMMPS molecular dynamics simulation engine[62,63]. Full computational details can be found in Supplementary Note 1.

### Electrode preparations
The electrodes were fabricated by an e-beam evaporator (Ulvac Inc.; deposition rate = 3 Å s⁻¹) for Ag and Au and by a sputter (Ulvac Inc.; deposition rate = 6 Å s⁻¹) for Cu at a vacuum level of $10^{-6}$–$10^{-7}$ Torr. Ag (99.99%), Au (99.99%), and Cu (99.99%) targets were deposited onto polytetrafluoroethylene (PTFE) membrane as a GDE with a pore size of 450 nm, and used as a cathode for CO₂ and/or CO electrolysis. Their successful preparations were investigated by the X-ray diffraction (XRD; MiniFlex 600, Rigaku) and scanning electron microscope (SEM; SU8230, Hitachi). The XRD pattern was obtained at a 40 kV accelerating voltage and a 15 mA current with a scan rate of 1° min⁻¹. The SEM and energy-dispersive X-ray spectroscopy (EDS) were taken at an acceleration voltage of 15.0 kV. The Nafion-coated Cu electrode was prepared by spraying a Nafion ink onto the Cu-PTFE electrode. The Nafion ink was prepared by mixing 2.5 mL of isopropanol and 30 μL of Nafion (5 wt%) solution, and its loading amount was set to 12.5 μg cm⁻².

NiNC catalyst was prepared from Ni$^{II}$ acetate tetrahydrate (98%, Sigma-Aldrich), phen (≥99%, Sigma-Aldrich), and ZIF-8 (ZnN₄C₈H₁₂, Basolite Z1200 from Sigma-Aldrich). The precursor mixture (1 g), containing 0.5 wt% Ni with a mass ratio phen/ZIF-8 of 20/80, was

homogenized by dry ball-milling (FRITSCH Pulverisette 7 Premium) for 4 cycles of 30 min at 400 rpm, and then pyrolyzed at 1050 °C in Ar (5 N, Daedeok) for 1 h. A $ZrO_2$ crucible with 100 zirconium oxide balls of 5 mm diameter was used for the ball-milling procedure. Atomically dispersed Ni sites stabilized upon N-doped carbon were characterized by the X-ray photoelectron spectroscopy (XPS), the extended X-ray absorption fine structure (EXAFS), and the high-angle annular dark-field scanning transmission electron microscopy (HAADF-STEM, FEI; Titan™ 80–300 TEM) equipped with a fast CCD camera (Gatan, One-view 1095). The XPS signal was collected with a Sigma Probe (Thermo VG Scientific) equipped with a micro-focused monochromator X-ray source. The EXAFS was collected in transmission mode at Pohang Accelerator Laboratory (7D-XAFS beamline) with an energy scale calibration by Ni foil. The NiNC electrode was prepared by spraying NiNC ink—4 mg catalyst, 200 μL Nafion solution (5 wt%), and 2800 μL IPA—onto a carbon paper ($1 \times 1$ cm$^2$ active area; TGP-H-090 with a 20 wt% PTFE content Toray) with a 1 mg cm$^{-2}$ NiNC. Prior to the NiNC electrode fabrication, hydrophobic mesoporous layer (MPL, preventing electrolyte leakage) was additionally introduced on the carbon paper by spraying a mixture of 100 mg Ketjen black EC-300J, 100 mg PTFE (60 wt%, Sigma-Aldrich), and 20 mL IPA (99.5%, Sigma-Aldrich) with 2 mg cm$^{-2}$ Ketjen black EC-300J loading, and subsequently by heat-treatments at 513 and 613 K under $N_2$ atmosphere for 30 min each.

## Electrochemical investigations

All electrochemical measurements were performed with a VMP-300 potentiostat (Bio-Logic). The $CO_2$ (4 N, Daedeok) and CO (4 N, Samjung) were electrolyzed in a home-made electrochemical flow cell (Supplementary Fig. 10)[6,64], in which a working electrode and a saturated Ag/AgCl reference electrode (RE-16, EC-Frontier) were physically separated from a Ni-foam counter electrode (MTI Korea) by an anion exchange membrane (AEM; fab-pk-130, Fumasep). Electrolytes were prepared using deionized water (≥18.2 MΩ, Arium® mini, Sartorius) with various chemicals (all supplied from Sigma-Aldrich): KOH (99.99%), $K_2CO_3$ (99.995%), $Li_2CO_3$ (99.999%), $Na_2CO_3$ (99.95–100.05%), $Rb_2CO_3$ (99.8%), $Cs_2CO_3$ (99.995%), LiOH (98%), NaOH (≥98%), RbOH (99.9%), CsOH (99.95%), $HClO_4$ (70 %,), $KH_2PO_4$ (99.0%), $H_3PO_4$ (85 wt%), $KHCO_3$ (99.95%), and NaF (>99%). The electrolytes continuously flowed into both anode and cathode compartments of the electrochemical cell with a flow rate of 5 mL min$^{-1}$. In the cathode compartment, $CO_2$ or CO gas flowed at the back of the working electrode at a flow rate of 20 mL min$^{-1}$. The reference electrode was calibrated against a Pt wire electrode (CE-1, EC-Frontier) in $H_2$-saturated electrolytes and converted to the RHE scale before every single measurement. The SHE and CCE scales were estimated by $E_{SHE} = E_{Ag/AgCl} + 0.197$ V and $E_{CCE} = E_{SHE} - 0.059 \times \log[M^+]$, respectively. The CO$_2$RR and CORR were conducted by a chronoamperometry (CA) for 1 h at each potential, and their polarization curves were manually IR-corrected (MIR, 85%). All gas products were analyzed using an online gas chromatograph (YL6500 GC, YL Instrument) equipped with a thermal conductivity detector (TCD) and a flame ionization detector (FID). A Carboxen-1000 column (12390-U, Supelco) was used for both TCD and FID, and Ar (99.999%) was used as a reference gas.

The $C_{diff}$ of Ag and Cu electrodes were measured using a conventional three-electrode system. A polycrystalline Ag (99.998%, Alfa Aesar) and Cu foils (99.99+%, Goodfellow), a graphite rod, and a saturated Ag/AgCl electrode were employed as the working, counter, and reference electrodes, respectively. Prior to every single measurement, the Ag electrode was chemically polished using the following procedure. The Ag electrode was first immersed in a solution mixture of 0.3 M KCN (≥96%, Sigma-Aldrich) and $H_2O_2$ (29–32%, Alfa Aesar) with a volume ratio of 1.5:1 for 3 s, during which gas was vigorously evolved, and thereafter it was exposed to air for another 3 s. The Ag electrode was subsequently soaked in a 0.55 M KCN solution until the

gas evolution ceased, and it was thoroughly washed with DI water. After repeating the chemical polishing procedure 10 times, a highly reflective surface was obtained. For the Cu electrode, it was polished mechanically with alumina slurry (1 and 0.05 μm, R&B Inc.) to remove native Cu oxide. The surface of the polished electrodes was protected by ultrapure water before it was transferred to the electrochemical cell. For measuring the $C_{diff}$ of Nafion-coated Cu, 5 wt% Nafion solution was drop-casted onto the Cu foil electrode with a target loading of 12.5 μg cm$^{-2}$, identical to $CO_2$ and CO electrolysis studies. The $C_{diff}$ was measured by SPEIS with a frequency of 20 Hz and a potential amplitude of 10 mV. The obtained impedance data were fitted by the RC circuit given as $Z = R + 1/i\omega C_{diff}$ (Supplementary Fig. 35)[17], where $R$ is the solution resistance, and $\omega$ is the circular frequency. The Ohmic loss was compensated during the SPEIS experiments. The $E_{PZC}$ was separately measured in a highly diluted 2 mM NaF solution and was defined as the potential where the smallest $C_{diff}$ value was observed.

## Data availability

All data is available in the main text or Supplementary Information. The main DFT data are available in the ioChem-BD[65] at https://doi.org/10.19061/iochem-bd-6-162. The main MD data and experimental data are available in the Zenodo at https://zenodo.org/badge/latestdoi/530912301.

## Code availability

The DFT-CES code has been deposited in the Github database without accession code at https://zenodo.org/badge/latestdoi/531050106[66].

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

## Acknowledgements

This research was supported by the Samsung Science and Technology Foundation under Project Number SSTF-BA2101-08, the National Research Foundation of Korea (NRF) grant funded by the Korea government (MSIT) (No. 2021R1A5A1030054), the KIST Institutional Program, and the Korea Institute of Science and Technology Information (KISTI) National Supercomputing Center with supercomputing resources including technical support (KSC-2020-CHA-0006). Experiments at PLS-II were supported in part by MSIT and POSTECH.

## Author contributions

S.-J.S. and H.C. contributed equally to this work, and mainly performed the DFT-CES simulation and electrochemical experiments, respectively. H.K. and C.H.C. conceived the initial idea and supervised the project. D.H.W. prepared electrodes. S.R., H.-S.O., D.H.K., D.-H.N., and T.L. contributed to part of the experimental and computational calculations. S.-J.S., H.C., H.K., and C.H.C. wrote the manuscript with contributions from all authors.

## Competing interests

The authors declare no competing interests.
