## [Peer review file · Nature Communications]

Reviewers' comments:

Reviewer #1 (Remarks to the Author):

I was eager to read this work, as lately “cation effect” in electrochemical CO₂ reduction (eCO₂R) has got back momentum, urging the need for a systematization of new disruptive concepts.

Monteiro et al. (ref 14) have demonstrated experimentally on gold, silver, and copper at acidic bulk pH that alkali cations enable eCO₂R by locally stabilizing CO₂ adsorption and facilitating the electron transfer (ET) to form the *CO₂- intermediate. A similar triggering effect was reported for K⁺ on a Cu gas diffusion electrode from neutral to alkaline surface pH (ref 3, Fig. S7). Variations in cation concentration at the outer Helmholtz plane are instead deemed responsible for activity trends, as first suggested by Resasco et al. (ref 25).

In this joint computational/experimental work, Shin et al. employ a first principles-based multiscale simulation to confirm steady coordination between cation and CO₂ for 5 ns (in ref 14 observed only for few ps). Consequently, they describe a cation-coupled electron transfer mechanism (almost equivalent to the scheme proposed in ref 14), which they demonstrate to be the rate-determining step for eCO₂R through kinetic studies. Finally, they attributed eCO₂R activity trends to variations in excess cation at the OHP, in line with refs. 14 and 25, and indicate ion-pairing interaction between cations and anions to influence cation accumulation.

As just discussed, the “cation effect” field is rapidly evolving and, in my opinion, future advances require either systematic assessment of previous disruptive works toward the definition of a general scheme (as, in principle, the cation-coupled electron transfer mechanism) or novel insights. Unfortunately, the current version of the manuscripts does not fulfill either of these scopes, since it appears a mere confirmation of earlier studies with few minor novel insights (electrode surface charge as proxy of local cation concentration, ion-pairing effects). Even though the paper is well-written, and experiments and simulations have been devised and carried out critically, I do not think the work deserves publication at this stage. If instead the authors are prone to tackle three relevant (and still open) questions, then I will be glad to assess a revised manuscript, which may have high impact in the field and thus be suitable for Nat. Commun.

1. The authors state that the CCET is the rate-determining step (RDS) for eCO₂R on Ag. Some recent studies disregarded this hypothesis, suggesting instead that *CO₂ adsorption is the RDS for eCO₂R on transition metals (Nat. Catal. 2021, 4, 1024–1031) since electron transfer to CO₂ is kinetically facile once the molecule is activated (J. Phys. Chem. C 2019, 123, 29278–29283). By comparing both hypotheses, I presume that cation is crucial in enabling CO₂ adsorption and then a facile ET follows. However, cation may as well further facilitate the electron transfer, thus having a two-fold promoting effect. I think that the authors should include an accurate estimation of kinetics of ET with and without cations (as in J. Phys. Chem. C 2019, 123, 29278–29283) and short discussion of Nat. Catal. 2021, 4, 1024–1031 and related works, highlighting the range of validity for both hypotheses (*CO₂—•••M⁺ vs *CO₂ as RDS).

2. Coming back to the range of validity, how universal is the observation of first-order kinetics for eCO₂RR on local cation concentration? Goyal and Koper (ref 35) reported variations in HER reaction order on cation concentration depending on bulk pH. Huang et al. (ref 3) observed that “K⁺ does not play a role in the activation of CO₂ nor does it suppress HER (from proton reduction) in a locally acidic environment.” At which surface (and/or bulk pH) do the authors expect the CCET scheme to break? Any quantitative experimental/computational results to support their answer? Is this scheme valid only for Ag or can it be extended to other metals? If generalization to other metals cannot be demonstrated, then I suggest including Ag in the title.

3. I find R3 (*COOH···M⁺ + e⁻ → *CO + M⁺ + OH⁻) in line 136 confusing. Do the authors suggest a cation effect also on this step, for instance promotion of the C-OH bond breaking? If this is the case, this should be properly discussed and explained in the manuscript. Otherwise, I suggest removing the cation in this line to keep the scheme consistent with the original one from ref 14.

Minor comments

- In my opinion, the introduction of the alkali metal cation activity-corrected electrode (ACE) scale is misleading for at least three reasons. First, the overall discussion is based on kinetics, while a change in potential scale suggests a cation effect on the overall thermodynamics. Bronsted–Evans–Polanyi relationships do exist, but thermodynamics and kinetics should not be mixed up. Second, cation effect depends on local cation concentration and not on bulk concentration. I agree that bulk and surface concentration are expected to correlate, but again I would keep the message qualitative to avoid future misunderstandings. Third, I do not understand why the authors use the term “activity” since it seems to me that they only refer to cation concentration here.

- Line 184. Resasco et al. employed thermodynamic insights to rationalize cation accumulation at the OHP before Monteiro et al. (see Fig. 11 in ref 25), thus their contribution should be mentioned first.

- Line 154. Is “staircase potentiometric electrochemical impedance spectroscopy” correct or should it be “staircase potential electrochemical impedance spectroscopy” instead?

- Line 37, Supplementary Information. Why were 8 excess electrons added to the cell in presence of a single CO₂ adsorbate? The authors should clarify the rationale behind this choice since it is not clear.

- Figures S17-S19 and Table S1 are not mentioned in the main text and as far as I know this does not fulfill Nature format requirements. I encourage the authors to mention them in line 227 (Methods section).

Reviewer #2 (Remarks to the Author):

In this work, the authors studied the cation effects on CO₂RR combining computational and experimental methods, and proposed that cation-coupled electron transfer (CCET) is a rate-determining

step in the CO₂RR. This work is well structured and provides useful insight into the mechanism of CO₂RR. Therefore, I would like to recommend its publication, but the following questions should be addressed before accepting for publication.

1. In their DFT-CES calculation, how the charge of the DFT part (electrode plus some surface species) is decided? In particular, at different potential conditions, the surface charges may vary, including the $^{*}\text{CO}_2^{\{\Delta - \}}$. A related question would be, what are the potential and pH conditions their models correspond to?
2. In calculating the CCET energetics, the authors stated that they use “the cation-coupled CO₂-adsorbate ($^{*}\text{CO}_2 - \dots \text{M}^+$) using the QM/MM total energy of the DFT-CES at the potential at point of zero charge...” . How the potential other than pzc, i.e. electric field in the EDL, would affect the total energy of coupled adsorbate?
3. All molecules are constrained in DFT-CES schemes, while the protonation of $^{*}\text{CO}_2^-$ to $^{*}\text{COOH}$ is exothermic in the CI-NEB calculation. The OH⁻ generated from deprotonation of water may react with bicarbonate. How may this affect the calculation and conclusion drawn from it? A related question is, in their calculation water serves as a proton donor, but would bicarbonate, being more acidic than water, be more likely to donate the proton?
4. Recently, there is some progress on ab initio molecular dynamics simulations of EDLs in literature, highlighting the importance of chemisorption of water (may also well be CO or CO₂ of relevance here) on EDL structures and capacitances. I suggest the authors to make a connection to those studies and discuss how the interfacial structures (surface intermediate with different coverages, ions, etc) may change with the electrochemical conditions and how they may affect the catalytic reactions such as CCET.

Reviewer #3 (Remarks to the Author):

In this study the authors employed theoretical calculations and electrochemical measurements to study the mechanism of alkali cation effect on CO₂ electroreduction. Alkali cations are found to coordinate to the reaction intermediates, and participate in the first electron transfer step, which is rate determining. Based on a linear correlation between CO current and surface charge density the authors propose that the cation effect is a local colligative property.

The finding of cation-coupled electron transfer as the rate determining step is interesting. However, it is not original since a similar theory has been put forward by Monteiro et al., as referenced in this manuscript. The second conclusion of this paper, that the cation effect is a local colligative property, is not supported by the experimental and theoretical results, and in fact contradicts previous findings. Especially the discussion about ion-pairing, in my opinion, is contradictory to the works the authors have

cited, as well as other reports. Although I find the authors ideas intriguing, currently I do not think the new insight proposed in this work is adequately supported. Consequently, I cannot recommend it for publication in Nature Communications.

Specific Comments:

1. The discussion about ion-pairing (line 200,201) contradicts the works the authors have cited and is somewhat misleading. Refs. 42-44 each claim that ion-pairing is favored by like-sized ions over ion pairs of dissimilar size. In the case of CO₂ electroreduction, where the anion is bicarbonate or carbonate, anions are even bigger than the largest cation, Cs⁺. Thus, larger cations like Cs⁺ should have a larger tendency than smaller cations to form a contact ion pairs. This point has also been confirmed by other experimental works [J. Chem., 2000, 53, 887–890]. The only reference, which is consistent with the claim that smaller cations demonstrate stronger ion-pairing (Ref. 41), is specifically conducted for proteins, which is quite different from the present case of electrocatalytic CO₂RR, and is not compelling in this context.

2. The mechanism of the cation effect (line 209-211), claiming that it originates predominantly from interfacial cation concentration rather than chemical interactions between cations and intermediates, is not sufficiently supported. Essentially, I disagree with the underlying assumption that chemical interactions in the EDL, which are responsible for CO₂RR activation, are not strongly influenced by cation composition.

For example, the authors do not show the calculated electrode potential or dipole moment for Rb⁺, which is actually very important since recent studies have shown Cs⁺ could break the cation-dependent trend in the interfacial solvation structure [JACS Au 2021, 1, 1674–1687], and Rb⁺ represents the turning point. Additionally, the authors should report the coordination numbers and O-M⁺ distance as they did for K⁺, for other cations to show whether these chemical interactions, which influence catalytic performance, are similar between cations (which I believe is not correct). Ideally the authors would also investigate cation concentration dependence in these calculations to show whether these important parameters are significantly impacted by cation density.

3. The alkali cations have very different solvation energies, hydrated radii, local solvation structures, polarizabilities, etc. These properties have been extensively investigated, both in the context of electrocatalytic interfaces and in solutions more generally. Here the authors show that CO₂RR activity is first order in cation concentration, leading to the conclusion that the cation effect is a colligative property. However, this conclusion is based on the assumption that specific interactions between cations and solvent as well as between cations and CO₂ is essentially unaffected by cation identity, which is almost certainly not correct. As explained in comment 2 above, this underlying assumption is first not specifically tested, and second contradicts numerous previous reports. This is my overarching

concern with the manuscript, and the primary reason I cannot recommend it for publication, despite the interesting correlation between CO₂RR activity and local cation concentration.

4. For the experimental evidence, the authors only perform kinetics measurements for different cations in 0.05 M carbonate solutions. More kinetics measurement in very diluted solutions (where interfacial cation concentration should be similar, not restricted by cation sizes) and very concentrated solutions (where interfacial cation concentration is different) [Energy Environ. Sci., 2019, 12, 3380--3389] should also be conducted for comparison.

5. The authors should provide the equivalent circuit they employed to fit the impedance result and explain how they derive the C_{diff} .

In summary, although the authors' findings show the importance of cation concentration for CO₂RR via a cation-coupled electron transfer mechanism, the main new conclusion of this work, that the cation effect is a colligative property driven by ion-pairing, is not supported and contradicts many previous experimental and theoretical findings. More careful investigations should be performed, specifically investigating differences in interfacial solvation structure between cations. Due to these concerns regarding both novelty and accuracy of the findings, I do not recommend this work for publication in Nature Communications.

Hyungjun Kim

Associate Professor

Department of Chemistry, E6-6 Building, Rm 506
291, Daehak-ro, Yuseong-gu, Daejeon
34141, Republic of Korea

KAIST

Korea Advanced Institute of
Science and Technology

Point-by-point responses to the reviewers

Comments to Reviewer. Thank you for your helpful comments, which are reproduced here in *italics*. Our responses are in **boldface**.

Reviewer #1 (Remarks to the Author):

I was eager to read this work, as lately “cation effect” in electrochemical CO₂ reduction (eCO₂R) has got back momentum, urging the need for a systematization of new disruptive concepts.

*Monteiro et al. (ref 14) have demonstrated experimentally on gold, silver, and copper at acidic bulk pH that alkali cations enable eCO₂R by locally stabilizing CO₂ adsorption and facilitating the electron transfer (ET) to form the *CO₂⁻ intermediate. A similar triggering effect was reported for K⁺ on a Cu gas diffusion electrode from neutral to alkaline surface pH (ref 3, Fig. S7). Variations in cation concentration at the outer Helmholtz plane are instead deemed responsible for activity trends, as first suggested by Resasco et al. (ref 25).*

In this joint computational/experimental work, Shin et al. employ a first principles-based multiscale simulation to confirm steady coordination between cation and CO₂ for 5 ns (in ref 14 observed only for few ps). Consequently, they describe a cation-coupled electron transfer mechanism (almost equivalent to the scheme proposed in ref 14), which they demonstrate to be the rate-determining step for eCO₂R through kinetic studies. Finally, they attributed eCO₂R activity trends to variations in excess cation at the OHP, in line with refs. 14 and 25, and indicate ion-pairing interaction between cations and anions to influence cation accumulation.

We appreciate the reviewer’s concise and clear summary of our work in the context of the field as a whole, their keen perspective, and valuable suggestions.

As just discussed, the “cation effect” field is rapidly evolving and, in my opinion, future advances require either systematic assessment of previous disruptive works toward the definition of a general scheme (as, in principle, the cation-coupled electron transfer mechanism) or novel insights. Unfortunately, the current version of the manuscripts does not fulfill either of these scopes, since it appears a mere confirmation of earlier studies with few minor novel insights (electrode surface charge as proxy of local cation concentration, ion-pairing effects). Even though the paper is well-written, and experiments and simulations have been devised and carried out critically, I do not think the work deserves publication at this stage. If instead the authors are prone to tackle three relevant (and still

Hyungjun Kim

Associate Professor

Department of Chemistry, E6-6 Building, Rm 506
291, Daehak-ro, Yuseong-gu, Daejeon
34141, Republic of Korea

KAIST

Korea Advanced Institute of
Science and Technology

open) questions, then I will be glad to assess a revised manuscript, which may have high impact in the field and thus be suitable for Nat. Commun.

We thank the reviewer for bringing up these three important questions in this field. By substantially expanding the scope of our work to identify CCET- and PCET-based mechanisms controlling various reaction paths of CO₂ electroreduction and unifying disruptive theoretical concepts about this unique phenomenon, our revised manuscript tackles key issues in the field – suitable for the high standard of Nature Communications.

*1. The authors state that the CCET is the rate-determining step (RDS) for eCO₂R on Ag. Some recent studies disregarded this hypothesis, suggesting instead that *CO₂ adsorption is the RDS for eCO₂R on transition metals [Nat. Catal. 2021, 4, 1024–1031] since electron transfer to CO₂ is kinetically facile once the molecule is activated [J. Phys. Chem. C 2019, 123, 29278–29283]. By comparing both hypotheses, I presume that cation is crucial in enabling CO₂ adsorption and then a facile ET follows. However, cation may as well further facilitate the electron transfer, thus having a two-fold promoting effect. I think that the authors should include an accurate estimation of kinetics of ET with and without cations [as in J. Phys. Chem. C 2019, 123, 29278–29283] and short discussion of Nat. Catal. 2021, 4, 1024–1031 and related works, highlighting the range of validity for both hypotheses (*CO₂⁻...M⁺ vs *CO₂ as RDS).*

We thank the reviewer for bringing up this issue. Consistent with the suggestion by the Chan group, we conceive that the electron transfer (ET) to CO₂ is kinetically facile once the molecule is activated, i.e., bent (the ET rate is estimated to be an order of 10¹⁵ s⁻¹). Once the CO₂ is bent, the ET can be considerably facilitated by the lowered LUMO of the CO₂, resulting in rapid ET. The bent CO₂ possesses radical character at the carbon center and can be stabilized when it is adsorbed on the metal surface, while cation coordination can enable further stability. Thus, from a mechanistic point of view, one can consider that the cation enables CO₂ adsorption –which accompanies a fast ET. The presence of a cation at the terminal of bent CO₂ not only chemically stabilizes the bent CO₂, but also downshifts the LUMO to also facilitate the ET, i.e., two-fold promoting effect, as mentioned by the reviewer (Figure R1).

Figure R1. Insight on the nature of cation-coupled electron transfer mechanism derived from the projected density of states (PDOS) analysis. **a**, A schematic showing the energy level change of CO₂, which is based on the PDOS analysis (Figure R1b). Before the adsorption, CO₂ at a linear form has 1π_g HOMO state and 2π_u LUMO state. When it is adsorbed to the metal, the states are broadened due to the mixing with metal states, and the bent form breaks the degeneracy of 2π_u and 1π_g states; one level shifts down from the 2π_u LUMO state and is partially filled, yielding a radical character. When a cation is coupled to the bent *CO₂, the LUMO state downshifts further due to the electric field from the cation. This increases the electronic occupation to the *CO₂, yielding an anionic character of *CO₂. E_f denotes the Fermi level. **b**, PDOS of CO₂ adsorbate (*CO₂) on Ag(111) calculated using DFT-CES. The Figures R1a and R1b are included in the “Figure 2f” and “Supplementary Figure 9” of the revised manuscript, respectively.

However, we note that the “adsorption-then-ET” mechanism predominantly reflects cause-and-effect type argument and not necessarily a temporal sequence. At this stage, one cannot distinguish between whether the fast ET will follow the adsorption over time in a sequential manner (similar to what is proposed by the Chan group) or whether it will be coupled with the adsorption step. Thus, we can consider two possible mechanisms that include cation coupling:

(mechanism 1)

(mechanism 2)

Hyungjun Kim

Associate Professor

Department of Chemistry, E6-6 Building, Rm 506
291, Daehak-ro, Yuseong-gu, Daejeon
34141, Republic of Korea

KAIST

Korea Advanced Institute of
Science and Technology

When the CO₂ is adsorbed on the catalyst surface, its electronic wavefunction is largely hybridized with the wavefunction of the catalyst surface. Thus, the first adsorption step of “mechanism 1” involves an electron redistribution between the adsorbate and the surface, which can be termed as an electronic polarization from electrode to adsorbate. Meanwhile, the first step of “mechanism 2” involves an ET, as specified in its reaction equation.

Strictly speaking, electrons exist as a cloud of indistinguishable quantum-mechanical particles. Consequently, the distinction between “polarization (mechanism 1)” and “transfer (mechanism 2)” depends on a hypothetical partitioning of the electron density in space, while both phenomena can be determined to be equivalent if there is a significant electronic overlap between the metal and the adsorbate [*J Mol Model* 2017, 23, 297]. Thus, we believe that there is no fundamental difference between “mechanism 1” and “mechanism 2”; they are two different vantage points of the same phenomenon. The former stems from a more continuum-level, electrostatic paradigm; the latter originates from a more atomic-level, charge-transfer paradigm. Such a duality of electronic polarization and electron transfer provides an important connection point between the “field-effect” scenario proposed by the Chan group and the “cation-coupled ET” scenario. This detailed discussion is provided on page 7 of the revised manuscript.

2. Coming back to the range of validity, how universal is the observation of first-order kinetics for eCO₂RR on local cation concentration? Goyal and Koper (ref 35) reported variations in HER reaction order on cation concentration depending on bulk pH. Huang et al. (ref 3) observed that “K⁺ does not play a role in the activation of CO₂ nor does it suppress HER (from proton reduction) in a locally acidic environment.” At which surface (and/or bulk pH) do the authors expect the CCET scheme to break? Any quantitative experimental/computational results to support their answer? Is this scheme valid only for Ag or can it be extended to other metals? If generalization to other metals cannot be demonstrated, then I suggest including Ag in the title.

We thank the reviewer for bringing up the issue of generalizing the CCET mechanism.

Prior to the detailed discussion about first-order kinetics of CO₂RR with respect to local cation concentration, we would like to first compare our findings with previous reports that apparently contradict the suggested CCET mechanism. In a previous work of Goyal and Koper [*Angew. Chem. Int. Ed.* 2021, 60, 13452–13462], the critical effects of alkali metal cations and electrolyte pHs on the activity of the hydrogen evolution reaction (HER) over the Au electrode were reported, revealing an increased HER activity with increasing bulk cation concentration and electrolyte pH. They highlighted that the promoting effects can be observed under limited conditions, *i.e.*, pH 11. In addition, Huang et al. reported that, in acidic environments, K⁺ does not play a specific role in CO₂RR [*Science* 2021, 372, 1074–1078]. These studies collectively imply that electrolyte

pH may be a critical parameter determining whether or not the alkali metal cation plays a key role in electrocatalysis.

Hence, we further investigated CO₂RR electrocatalysis on Ag electrode in various electrolyte pHs, ranging from highly acidic (pH 1.6), to nearly neutral (pH 6.8), and highly alkaline (pH 15) conditions. The results are summarized in Figure R2; the polarization curves were plotted against SHE, RHE, and ACE reference potential scales. The results clearly demonstrate a clear collapse of the polarization curves on the ACE scale at an electrolyte pH higher than 8.6, while non-negligible deviations were observed at pH lower than 6.8, implying that CO₂RR on Ag was primarily governed by the CCET pathway in alkaline conditions.

Figure R2. CO₂-to-CO conversion on the Ag electrode in acidic, neutral, and alkaline electrolytes. The j_{CO} vs. potential curves measured on the Ag electrode in an electrochemical flow cell. The CO₂RR polarization curves are plotted with respect to a, SHE, b, RHE ($V_{\text{SHE}} - 0.059 \times \log[\text{H}^+]$), or c, ACE ($V_{\text{SHE}} - 0.059 \times \log[\text{K}^+]$) scales, respectively. Electrolytes were 0.03 M HClO₄ + 0.07 M KClO₄ for pH 1.6, 0.03 M H₃PO₄ + 0.07 M KH₂PO₄ M for pH 3, 0.1 M KH₂PO₄ for pH 4.3, CO₂-saturated 0.1 M KHCO₃ for pH 6.8, Ar-saturated 1.5 M KHCO₃ for pH 8.6, 0.05 M K₂CO₃ for pH 11, 0.01–10 M KOH for pH 12–15. The results reveal that clear collapse of the polarization curves upon the ACE scale is limitedly found at pH higher than 8.6, indicating that the CCET may hardly govern the RDS of CO₂RR in neutral and acidic environments. This is included in the “Supplementary Figure 16” of the revised manuscript.

We subsequently investigated CO₂RR with various electrode materials to identify whether the CCET mechanism is only achievable on the Ag electrode or also on other CO₂RR-active materials. Metallic Au (Figure R3) and single-atom NiNC (Figure R4) electrodes (well-known electrodes which produce CO with high selectivity) were studied. Similar to the Ag case, the results demonstrated the collapse of j_{CO} curves on the ACE scale for both Au and NiNC electrodes. We also notably identified the collapse of polarization curves for the CO-to-C₂H₄ conversion on the Cu electrode (Figure R5). The latter finding motivated significant revisions to the manuscript to merge the overall mechanistic scheme of CO₂RR on the Cu electrode with identifying RDS for

CO, CH₄, and C₂H₄ products. Following additional computational and experimental efforts, we demonstrated that the CCET is not a singular event on the Ag electrode but can also be found in other representative electrodes for CO₂RR.

Figure R3. CO₂-to-CO conversion on the Au electrode. The j_{CO} vs. potential curves measured on the Au electrode in an electrochemical flow cell. The electrolytes were a–c, 0.01–10 M KOH and d–f, 0.01 M KOH + 0, 0.045, and 0.495 M K₂CO₃. The polarization curves are plotted with respect to the a,d, SHE, b,e, RHE ($V_{\text{SHE}} - 0.059 \times \log[\text{H}^+]$), or c,f, ACE ($V_{\text{SHE}} - 0.059 \times \log[\text{K}^+]$) scales. The grey lines indicate a Tafel slope (typically plotted as an inverse function of the present polarization curve) of 120 mV dec⁻¹. A collapse of the polarization curves is found upon the ACE scale, highlighted with a red box. This is included in the “Supplementary Figure 18” of the revised manuscript.

Figure R4. CO₂-to-CO conversion on the NiNC electrode. The j_{CO} vs. potential curves measured on the NiNC electrode in an electrochemical flow cell. The electrolytes were a–c, 0.01–10 M KOH and d–f, 0.01 M KOH + 0, 0.045, and 0.495 M K₂CO₃. The polarization curves are plotted with respect to the a,d, SHE, b,e, RHE ($V_{\text{SHE}} - 0.059 \times \log[\text{H}^+]$), or c,f ACE ($V_{\text{SHE}} - 0.059 \times \log[\text{K}^+]$) scales. The grey lines indicate a Tafel slope (typically plotted as an inverse function of the present polarization curve) of 120 mV dec⁻¹. A collapse of the polarization curves is found upon the ACE scale, highlighted with a red box. This is included in the “Supplementary Figure 20” of the revised manuscript.

Figure R5. Electrochemical CORR for the CO-to-C₂H₄ conversions measured on the Cu electrode in 0.05–5 M KOH electrolytes. All the electrolysis measurements were performed in an electrochemical flow cell. The polarization curves are plotted with respect to the a, SHE, b, RHE ($V_{\text{SHE}} - 0.059 \times \log[\text{H}^+]$), or c, ACE ($V_{\text{SHE}} - 0.059 \times \log[\text{K}^+]$) scales. The grey lines indicate a Tafel

slope (typically plotted as an inverse function of the present polarization curve) of 120 mV dec^{-1} . Collapses of the polarization curves are found for the C_2H_4 formations upon the ACE scale, which are highlighted with red boxes. This is included in the “Figure 3” of the revised manuscript.

3. I find R3 ($*\text{COOH}\cdots\text{M}^+ + \text{e}^- \rightarrow * \text{CO} + \text{M}^+ + \text{OH}^-$) in line 136 confusing. Do the authors suggest a cation effect also on this step, for instance promotion of the C-OH bond breaking? If this is the case, this should be properly discussed and explained in the manuscript. Otherwise, I suggest removing the cation in this line to keep the scheme consistent with the original one from ref 14.

We thank the reviewer and agree that the R3 reaction step ($*\text{COOH}\cdots\text{M}^+ + \text{e}^- \rightarrow * \text{CO} + \text{M}^+ + \text{OH}^-$) in the original manuscript can be misleading. Implementing the reviewer’s suggestion, additives for reaction intermediate (e.g., $\text{M}^+ + \text{OH}^-$) have been deleted. Instead, we now provide full reaction pathways for CO_2 -to- CO , CO -to- CH_4 , and CO -to- C_2H_4 (Figures R6 and R7). In the reaction pathways, cation-coordinated intermediates, classified by the DFT-CES simulation (Figure R8), are specified by appending $\cdots\text{M}^+$.

Figure R6. Reaction free energy (ΔG) diagrams for, a, CO_2 -to- CO on Ag(111), b, CO -to- CH_4 on Cu(100), and, c, CO -to- C_2H_4 on Cu(100). Applied potentials are -0.5 and $-1 \text{ V}_{\text{SHE}}$. The asterisk (*) indicates the adsorbed state, and the energetically favorable paths among other possibilities (Figure R7) are selectively shown. This is included in the “Figure 2” of the revised manuscript.

Figure R7. Reaction energy diagram for CO-to-CH₄ and CO-to-C₂H₄ reaction paths on Cu(100), calculated using DFT-CES energetics. The reaction free energy (ΔG) is calculated for different reaction paths. The cation-coordinated intermediates are specified by appending $\cdots M^+$. a,b, Reaction energy diagrams for the CO-to-CH₄ reaction path at $-0.5 V_{SHE}$ (a) and those at $-1.0 V_{SHE}$ (b). c,d, Reaction energy diagrams for the CO-to-C₂H₄ reaction path at $-0.5 V_{SHE}$ (c) and those at $-1.0 V_{SHE}$ (d). Energetically feasible reaction paths are highlighted using black solid line, while the other ones are shown in red. This is included in the “Supplementary Figure 6” of the revised manuscript.

Figure R8. Map of cation-(un)coordinated intermediates during CO₂RR. a, Temporal change of distance (r) between intermediate and K⁺ during 5-ns DFT-CES trajectory (grey line). Applied potential is $-1.0 \text{ V}_{\text{SHE}}$. The black solid line denotes a moving average of the grey line using a 0.05-ns time window. b–d, Possible intermediates for CO₂RR, where cation-coordinated species are identified. Reaction paths are denoted using different background colors; blue for CO₂-to-CO (b), green for CO-to-CH₄ (c), and red for CO-to-C₂H₄ (d). Cation CN to each intermediate is provided in Figure R9, and the species with CN ~ 1 are identified as cation-coordinated intermediates. All cartooned structures are based on the DFT-CES simulation (Figure R9). The dark-grey, red, white, and purple colored ball indicates the C, O, H, and K⁺, respectively, and the electrode is shown with the hatch patterned box. This is included in the “Figure 1” of the revised manuscript.

Figure R9. Radial distribution function, $g(r)$, calculated using the DFT-CES at -1.0 V_{SHE}. A radial distance, r , is defined between K^+ and O of adsorbates. When the adsorbates have no O, C is used to define r . The integrated (Int.) value of $g(r)$, $\int_0^r 4\pi r'^2 g(r') dr'$, shows the K^+ coordination number (CN) to the adsorbates. The adsorbates are classified into two groups; a, ones coordinated by a cation with CN ~ 1 , and b, the other ones that are not coordinated by a cation, where CN < 1 . This is included in the “Supplementary Figure 4” of the revised manuscript.

Minor comments

- In my opinion, the introduction of the alkali metal cation activity-corrected electrode (ACE) scale is misleading for at least three reasons. First, the overall discussion is based on kinetics, while a change in potential scale suggests a cation effect on the overall thermodynamics. Bronsted–Evans–Polanyi relationships do exist, but thermodynamics and kinetics should not be mixed up. Second, cation effect depends on local cation concentration and not on bulk concentration. I agree that bulk and surface concentration are expected to correlate, but again I would keep the message qualitative to avoid future misunderstandings. Third, I do not understand why the authors use the term “activity” since it seems to me that they only refer to cation concentration here.

We thank the reviewer to point this issue out. The collapse of polarization curves on the ACE scale is a thermodynamic observation, while the CCET mechanism should be understood from kinetic point of view. We broke down these two dimensions into separate sections titled “Cation concentration dependent Nernstian shifts” and “Cation effect as a local colligative property”, respectively. As noted by the reviewer, although the Bell-Evans-Polanyi principle furnishes a model describing the intimate relationship between the thermodynamics and kinetics [*Proc. R. Soc. Lond. A* 1936, 154, 414–429 and *Trans. Faraday Soc.* 1936, 33, 448–452], these two terms are not ultimately interchangeable, and their discussions need to be appropriately separated. Therefore, while the collapse of CO₂RR (and CORR) polarization curves on the ACE scale is interesting, it is unclear whether it fundamentally supports our argument that RDSs of the CO and C₂H₄ paths involve the CCET step.

The polarization curves represent kinetics, typically described by the Butler-Volmer equation. Whereas the ACE scale is defined here as “ $E_{ACE} = E_{SHE} - 0.059 \times \log[M^+]$ ”, reflecting a compensation of the potential contribution of alkali metal cation concentration (according to the thermodynamic Nernst equation). In a multi-electron (n) transfer process, $O + ne^- \rightleftharpoons R$, with an RDS of the first one-electron transfer step ($O + e^- \rightleftharpoons R'$) possesses a current–potential characteristic given by

$$i = -nFAk_{RDS}^0 \left[C_O(0) \exp \left\{ -\frac{\alpha_c F}{RT} (E - E_{RDS}^{0'}) \right\} - C_{R'}(0) \exp \left\{ \frac{\alpha_a F}{RT} (E - E_{RDS}^{0'}) \right\} \right] \quad (\text{Eq. R1})$$

where A is the electrode area, k_{RDS}^0 is the standard rate constant for the RDS step, $C_O(0)$ (and $C_{R'}(0)$) is the concentration of species O (and R') at the electrode surface, α_c (and α_a) is the cathodic (and anodic) charge transfer coefficient, and $E_{RDS}^{0'}$ is the formal potential of the RDS step [Kinetics of electrode reactions. In *Electrochemical methods: fundamentals and applications*. John Wiley & Sons, Inc., 2001]. At high cathodic overpotential (η), the current-potential characteristic can be rewritten as the following equation.

$$i = -nFAk_{RDS}^0 \left[C_O(0) \exp \left\{ -\frac{\alpha_c F}{RT} (E - E_{RDS}^{0'}) \right\} \right] \quad (\text{Eq. R2})$$

Supposing the surface concentration of O, $C_O(0)$, depends on its bulk concentration (C_O^*) via $C_O(0) = \beta C_O^{*\gamma}$, where β is a proportionality coefficient, and γ is an exponent, then (Eq. R2) can be written in terms of C_O^* ;

$$\begin{aligned} i &= -nFAk_{RDS}^0 \left[\beta C_O^{*\gamma} \exp \left\{ -\frac{\alpha_c F}{RT} (E - E_{RDS}^{0'}) \right\} \right] \\ &= -nFAk_{RDS}^0 \left[\beta \exp \left[-\frac{\alpha_c F}{RT} \left\{ E - \left(E_{RDS}^{0'} + \frac{\gamma RT}{\alpha_c F} \ln C_O^* \right) \right\} \right] \right] \end{aligned}$$

which is further reduced to the following form at T = 25 °C;

Hyungjun Kim

Associate Professor

Department of Chemistry, E6-6 Building, Rm 506
291, Daehak-ro, Yuseong-gu, Daejeon
34141, Republic of Korea

KAIST

Korea Advanced Institute of
Science and Technology

$$i = -nFAk_{\text{RDS}}^0 \left[\beta \exp \left[-\frac{\alpha_c F}{RT} \left\{ E - (E_{\text{RDS}}^{0'} + 0.059 \frac{\gamma}{\alpha_c} \log C_0^*) \right\} \right] \right] \quad (\text{Eq. R3})$$

Here, the last term, $E_{\text{RDS}}^{0'} + 0.059 \frac{\gamma}{\alpha_c} \log C_0^*$, is dictated by a shift of the formal potential of the RDS step by $0.059 \frac{\gamma}{\alpha_c}$ V per $\log C_0^*$. The special case of $\gamma = \alpha_c$, (Eq. R3) further indicates the same kinetics upon a so-called Nernstian shift of the potential.

The α_c is generally 0.5, while the Gouy-Chapman theory also predicts the γ to be 0.5 [Structure of the electric double layer. In *Electrochemical systems*. John Wiley & Sons, Inc., 2004]. In addition, our experimental results, as shown in Fig. 4a (here, Figure R10a), demonstrate the $|\sigma|$ to be proportional to the square root of the bulk ion concentration – consistent with $\gamma = 0.5$. Therefore, the $\gamma = \alpha_c$ relation is valid for the cases studied here, clearly demonstrating how the thermodynamics and kinetics are interrelated in CO₂RR (and CORR), while supporting the validity of our data interpretation on the ACE scale.

These are discussed in the “Supplementary Note 3” of the revised manuscript.

The second point pertains to the observation above that the local cation concentration is connected with the cation effect, not the bulk cation concentration. As discussed above, we demonstrated the correlation between the bulk and local cation concentrations to identify cation effect on CO or C₂H₄ production as a local colligative property (Figure R10). Furthermore, the rationale of empirically-determined conditions that maximize C₂H₄ production is elucidated in the context of our mechanism (light blue points in Figure R10c and Figure R11).

Figure R10. Reaction kinetic studies. a, Correlation plots between bulk cation concentration ($[M^+]$) and surface charge density ($|\sigma|$) for the Ag and Cu electrodes. Correlation plots of b, j_{CO} on the Ag electrode and c, $j_{\text{C}_2\text{H}_4}$ on the Cu electrode with $|\sigma|$. The data was collected at $-1.3 V_{\text{SHE}}$

for the Ag electrode (CO_2 -to- CO formation) and $-1.4 \text{ V}_{\text{SHE}}$ for the Cu electrode (CO -to- C_2H_4 formation). Electrolytes used for gathering these data can be classified into five different categories: KOH electrolytes with different concentrations (filled circles), 0.01 M KOH electrolytes with additional K_2CO_3 salt (double-crossed circles), 0.5 M KOH electrolytes with additional K_2CO_3 salt (single-crossed circles), 0.05 M M_2CO_3 electrolytes (half-filled symbols), and 1 M MOH electrolytes (filled symbols), where M is Li, Na, K, Rb, and Cs. In addition, a result collected on the Nafion-coated Cu electrode in 5 M KOH electrolyte (a filled circle with pale blue color) was also provided. Here, the total M^+ concentration and its identity are distinguished by color and symbol shape, respectively. Guidelines for the slopes are indicated by a dashed line. This is included in “Figure 4” of the revised manuscript.

Figure R11. Boosted C_2H_4 formation on the Nafion-coated Cu electrode. a, The $j_{\text{C}_2\text{H}_4}$ vs. potential curves and b, the C_{diff} curves, measured on bare and Nafion-coated Cu electrodes in 5 M KOH electrolyte. c, Schematic descriptions of the catalyst–electrolyte interface for the Nafion-free and Nafion-coated Cu electrodes, showing enriched population of M^+ near the interface on the Nafion-coated one. This is included in the “Figure 5” of the revised manuscript.

Hyungjun Kim

Associate Professor

Department of Chemistry, E6-6 Building, Rm 506
291, Daehak-ro, Yuseong-gu, Daejeon
34141, Republic of Korea

KAIST

Korea Advanced Institute of
Science and Technology

Finally, we revised the term from the “alkali metal cation activity-corrected electrode” to the “alkali metal cation concentration-corrected electrode” – consistent with the reviewer’s recommendation.

- Line 184. Resasco et al. employed thermodynamic insights to rationalize cation accumulation at the OHP before Monteiro et al. (see Fig. 11 in ref 25), thus their contribution should be mentioned first.

We thank the reviewer and agree that the sentence in the original manuscript can be misleading to some readers. In the revised manuscript, the study from Resasco et al. is now explicitly referenced first on page 10 (pasted below).

“Instead, reaction kinetic study, which can provide definite evidence on reaction mechanism⁴⁸, unravels that different CO or C₂H₄ production rates depending on the M⁺ identity (and its bulk concentration) are primarily attributed to different local cation concentrations at the interface, supporting the previous computation-based claim^{16,17}”

Here, ref 17 is the paper from Resasco et al.

- Line 154. Is “staircase potentio electrochemical impedance spectroscopy” correct or should it be “staircase potential electrochemical impedance spectroscopy” instead?

We thank the reviewer and, here, potentio refers to the potential; however, the conventional terminology for this measurement has been coined as “staircase potentio electrochemical impedance spectroscopy”.

- Line 37, Supplementary Information. Why were 8 excess electrons added to the cell in presence of a single CO₂ adsorbate? The authors should clarify the rationale behind this choice since it is not clear.

Since CO₂ does not bind to the surface at the point of zero charge (PZC), we optimized the geometry of *CO₂ using the negatively charged surface. We found that the CO₂ binds to the surface when the number of excess electrons exceeds 2, and the original choice of 8 was simply to assure the surface to be large enough to be charged. To check whether such an arbitrary choice could alter the key results, we compared the simulation results using the *CO₂ geometry

optimized on the surface with 8 excess electrons (Geometry 1; Figure R12), and those using the *CO₂ geometry optimized on the surface with 3 excess electrons (Geometry 2; Figure R13), both of which are essentially the same.

Geometry 1 (manuscript)

Figure R12. Summary of *CO₂ on Ag(111) DFT-CES results using the *CO₂ geometry optimized on the surface with 8 excess electrons (geometry 1). **a**, Averaged distance between the two O atoms of *CO₂ and K⁺ is plotted as a function of the simulation time in a grey color. The black solid line denotes a moving average of the grey line using a 0.05-ns time window. **b**, The radial distribution function, $g(r)$, and its integrated (Int.) value, $\int_0^r 4\pi r'^2 g(r') dr'$, is plotted as a function of radial distance between O of *CO₂ and K⁺. The Int. $g(r)$ shows the K⁺ coordination number to the

adsorbates. **c**, Projected density of states (PDOS) of $^*\text{CO}_2$. E_f denotes the Fermi level. **d**, Reaction free energy (ΔG) diagrams for CO_2 -to- CO . The all data here are included in the manuscripts (Figures 1a and 2a) and Supplementary Information (Supplementary Figures 3, 4, 5, and 9).

Figure R13. Summary of $^*\text{CO}_2$ on Ag(111) DFT-CES results using the $^*\text{CO}_2$ geometry optimized on the surface with 3 excess electrons (geometry 2). a, Averaged distance between the two O atoms of $^*\text{CO}_2$ and K^+ is plotted as a function of the simulation time in a grey color. The black solid line denotes a moving average of the grey line using a 0.05-ns time window. b, The radial distribution

Hyungjun Kim

Associate Professor

Department of Chemistry, E6-6 Building, Rm 506
291, Daehak-ro, Yuseong-gu, Daejeon
34141, Republic of Korea

KAIST

Korea Advanced Institute of
Science and Technology

function, $g(r)$, and its integrated (Int.) value, $\int_0^r 4\pi r'^2 g(r') dr'$, is plotted as a function of radial distance between O of *CO_2 and K^+ . The Int. $g(r)$ shows the K^+ coordination number to the adsorbates. c, Projected density of states (PDOS) of *CO_2 . E_f denotes the Fermi level. d, Reaction free energy (ΔG) diagrams for CO_2 -to- CO .

- Figures S17-S19 and Table S1 are not mentioned in the main text and as far as I know this does not fulfill Nature format requirements. I encourage the authors to mention them in line 227 (Methods section).

We thank the reviewer to point this out. We checked all the figures were properly called in the revised manuscript.

Reviewer #2 (Remarks to the Author):

In this work, the authors studied the cation effects on CO₂RR combining computational and experimental methods, and proposed that cation-coupled electron transfer (CCET) is a rate-determining step in the CO₂RR. This work is well structured and provides useful insight into the mechanism of CO₂RR. Therefore, I would like to recommend its publication, but the following questions should be addressed before accepting for publication.

We appreciate the reviewer's clear summary of our work. We fully appreciate the reviewer's valuable suggestions and have endeavored to address them comprehensively.

*1. In their DFT-CES calculation, how the charge of the DFT part (electrode plus some surface species) is decided? In particular, at different potential conditions, the surface charges may vary, including the *CO₂⁻. A related question would be, what are the potential and pH conditions their models correspond to?*

We thank the reviewer to point this out. In this study, two different charge states are employed: one is at a neutral condition and the other one possesses $-18 \mu\text{C cm}^{-2}$ surface charge density, yielding the electrode potential of $-0.5 \text{ V}_{\text{SHE}}$ and $-1.0 \text{ V}_{\text{SHE}}$, respectively, for both Ag and Cu electrodes (Figure R14). Also, we considered the pH condition of 12 to construct the reaction free energies.

Figure R14. Electrostatic potential profiles, ϕ , across the charged interfaces, calculated using DFT-CES. Two different surface charge density (σ) conditions, $\sigma = 0$ for the point of zero charge (PZC), and $\sigma = -18 \mu\text{C cm}^{-2}$, are compared for a, Ag(111) electrode and b, Cu(100) electrode. The ϕ is aligned to set the Fermi level at $\phi = 0$ V. By subtracting the 4.44 V from the calculated absolute electrode potential values [*Electrochim. Acta* 1990, 35, 269–271], the electrode potential vs. standard hydrogen electrode (SHE) are calculated, which are $-0.5 \text{ V}_{\text{SHE}}$ at PZC and $-1.0 \text{ V}_{\text{SHE}}$ at $\sigma = -18 \mu\text{C cm}^{-2}$ for both Ag(111) and Cu(100) electrodes. This is included in the “Supplementary Figure 2” of the revised manuscript.

2. In calculating the CCET energetics, the authors stated that they use “the cation-coupled CO_2^- adsorbate ($^*\text{CO}_2^- \cdots \text{M}^+$) using the QM/MM total energy of the DFT-CES at the potential at point of zero charge...”. How the potential other than pzc, i.e. electric field in the EDL, would affect the total energy of coupled adsorbate?

We thank the reviewer to suggest this point. We have implemented the reviewer’s comment and reaction paths calculated from the DFT-CES have now been evaluated not only at the PZC ($-0.5 \text{ V}_{\text{SHE}}$), but also at a more negative potential ($-1.0 \text{ V}_{\text{SHE}}$) in Figures R15 and R16. At the more negative potential condition, the electric field in the EDL is naturally included in the total energy expression of the DFT-CES, as illustrated by the reduced energy barriers in comparison with the PZC condition (Figure R15).

Figure R15. Reaction free energy (ΔG) diagrams for, a, CO_2 -to- CO on $\text{Ag}(111)$, b, CO -to- CH_4 on $\text{Cu}(100)$, and, c, CO -to- C_2H_4 on $\text{Cu}(100)$. Applied potentials are -0.5 and -1 V_{SHE} . The asterisk (*) indicates the adsorbed state, and the energetically favorable paths among other possibilities (Figure R16) are selectively shown. This is included in the “Figure 2” of the revised manuscript.

Figure R16. Reaction energy diagram for CO -to- CH_4 and CO -to- C_2H_4 reaction paths on $\text{Cu}(100)$, calculated using DFT-CES energetics. The reaction free energy (ΔG) is calculated for different reaction paths. The cation-coordinated intermediates are specified by appending $\cdots\text{M}^+$. a,b, Reaction energy diagrams for the CO -to- CH_4 reaction path at -0.5 V_{SHE} (a) and those at -1.0 V_{SHE} (b). c,d, Reaction energy diagrams for the CO -to- C_2H_4 reaction path at -0.5 V_{SHE} (c) and those at -1.0 V_{SHE} (d). Energetically feasible reaction paths are highlighted using black solid line, while the other ones are shown in red. This is included in the “Supplementary Figure 6” of the revised manuscript.

Hyungjun Kim

Associate Professor

Department of Chemistry, E6-6 Building, Rm 506
291, Daehak-ro, Yuseong-gu, Daejeon
34141, Republic of Korea

KAIST

Korea Advanced Institute of
Science and Technology

3. All molecules are constrained in DFT-CES schemes, while the protonation of $^*CO_2^-$ to *COOH is exothermic in the CI-NEB calculation. The OH- generated from deprotonation of water may react with bicarbonate. How may this affect the calculation and conclusion drawn from it? A related question is, in their calculation water serves as a proton donor, but would bicarbonate, being more acidic than water, be more likely to donate the proton?

We thank the reviewer for bringing up this interesting issue. However, while extensively revising our manuscript, we decided to focus on the thermodynamic barrier (as determined by the energetic difference among possible intermediates) rather than investigating the transition states; we thus removed our CI-NEB calculation results. We believe this is reasonable in terms of maintaining the consistency of the calculation level. Instead, as shown in Figures R15 and R16 (in response to comment 2), energy diagrams for the CO, CH₄, and C₂H₄ formation paths have now been included. Therefore, we reserve the investigation of the effect of proton donor species on the barrier of the protonation step for a future study.

4. Recently, there is some progress on *ab initio* molecular dynamics simulations of EDLs in literature, highlighting the importance of chemisorption of water (may also well be CO or CO₂ of relevance here) on EDL structures and capacitances. I suggest the authors to make a connection to those studies and discuss how the interfacial structures (surface intermediate with different coverages, ions, etc) may change with the electrochemical conditions and how they may affect the catalytic reactions such as CCET.

We thank the reviewer to make this suggestion. Following the reviewer's important comment, we highlighted the importance of recent computational progresses on page 4 of the revised manuscript – elucidating the EDL structural details as follows:

“Recent advances in computational simulations enable a direct investigation of the electrode-electrolyte structure, highlighting the importance of the atomic arrangement of EDL constituents (e.g., chemisorbed water, Helmholtz ions, etc.) at the buried nanoscale region, which can affect the catalytic reactions under the actual electrochemical conditions [*Phys. Rev. Lett.* 2017, 119, 016801, *J. Phys. Chem. Lett.* 2021, 12, 7299–7304, *Sci. Adv.* 2020, 6, eabb1219, *J. Chem. Phys.* 2018, 149, 084705, *J. Catal.* 2021, 396, 251–260, *Cell Rep. Phys. Sci.* 2022, 3, 100759, *Nat. Commun.* 2022, 13, 174, *ACS Catal.* 2018, 8, 2420–2427].”

Hyungjun Kim

Associate Professor

Department of Chemistry, E6-6 Building, Rm 506
291, Daehak-ro, Yuseong-gu, Daejeon
34141, Republic of Korea

KAIST

Korea Advanced Institute of
Science and Technology

Reviewer #3 (Remarks to the Author):

In this study the authors employed theoretical calculations and electrochemical measurements to study the mechanism of alkali cation effect on CO₂ electroreduction. Alkali cations are found to coordinate to the reaction intermediates, and participate in the first electron transfer step, which is rate determining. Based on a linear correlation between CO current and surface charge density the authors propose that the cation effect is a local colligative property.

We thank the reviewer for clearly summarizing key findings of our work.

The finding of cation-coupled electron transfer as the rate determining step is interesting. However, it is not original since a similar theory has been put forward by Monteiro et al., as referenced in this manuscript. The second conclusion of this paper, that the cation effect is a local colligative property, is not supported by the experimental and theoretical results, and in fact contradicts previous findings. Especially the discussion about ion-pairing, in my opinion, is contradictory to the works the authors have cited, as well as other reports. Although I find the authors ideas intriguing, currently I do not think the new insight proposed in this work is adequately supported. Consequently, I cannot recommend it for publication in Nature Communications.

We are grateful for the reviewer's keen point. We believe that our previous manuscript, where the cation effect only on the CO₂-to-CO production was examined, was *per se* distinct from the work of the Koper group in that we experimentally demonstrated (based on a kinetic study) the most direct evidence for the coupling of cations in the RDS of CO₂ reduction to CO. However, to further enhance the impact of our findings, we substantially revised (almost rewrote) the manuscript. The revised manuscript reports novel conclusions beyond the previous work by Monteiro et al., which can be summarized as follows:

- (1) We deconstructed the CCET-based mechanisms for not only the CO₂-to-CO formation path but also C₂ formation path to include the C-C coupling step, which is a key elementary step to produce value-added products such as C₂H₄. Moreover, we identify CCET- and PCET-based reaction paths over the course of multiple electron transfer steps of CO₂RR.**
- (2) We established a mechanistic link between the CCET-based RDS and the cation-species-dependent activity of CO₂RR, often referred to as a “cation effect”.**
- (3) We theoretically unified the previous “field effect” and “cation coupling” scenarios. The so-called “field” can be identified as one generated by the “coupled cation”, which is a macroscopically conceptualized atomic-scale realization of the Helmholtz plane. It not only**

Hyungjun Kim

Associate Professor

Department of Chemistry, E6-6 Building, Rm 506
291, Daehak-ro, Yuseong-gu, Daejeon
34141, Republic of Korea

KAIST

Korea Advanced Institute of
Science and Technology

stabilizes the adsorbate, but also facilitates electron transfer, both of which form the essence of the field effect.

(4) We elucidated the origin of empirical findings to maximize C₂H₄ production in the context of the CCET-based mechanism.

Thus, we believe that these findings have a tremendous impact on enhancing our understanding of the cation effect and the inherent chemical nature of the CCET, which is a rapidly evolving field still requiring a number of advances to build a generally applicable framework.

Regarding the reviewer's second point about the local colligative property, we note that our finding of local colligativity is not derived from an *assumption* that the interaction strengths of different cation identities are the same, but is derived from an experimental *observation* that the catalytic activity is singly correlated with the local cation concentration regardless of the identity of the cation species. This experimental observation dictates that the local concentration is a dominant factor leveling the effect of different interaction strengths of different cations (which are otherwise not the same). We argue that all kinds of colligative properties, such as a boiling point elevation and an osmotic pressure, can be observed in conjunction with various ionic species even though the magnitude of specific interactions between ions and water vary significantly.

However, we concede that our previous interpretation based on ion-pairing trends to explain this experimental observation was not sufficiently cogent; this interpretation has been removed from the revised manuscript. Instead, by substantially expanding the scope of our work to identify CCET- and PCET-based mechanisms controlling the various reaction paths of CO₂ electroreduction, while also unifying disruptive theoretical concepts to explain this unique phenomenon, our revised manuscript now tackles key issues in the field in a manner compatible with the high standard of the Nature Communications.

Specific Comments:

1. The discussion about ion-pairing (line 200,201) contradicts the works the authors have cited and is somewhat misleading. Refs. 42-44 each claim that ion-pairing is favored by like-sized ions over ion pairs of dissimilar size. In the case of CO₂ electroreduction, where the anion is bicarbonate or carbonate, anions are even bigger than the largest cation, Cs⁺. Thus, larger cations like Cs⁺ should have a larger tendency than smaller cations to form a contact ion pairs. This point has also been confirmed by other experimental works [J. Chem., 2000, 53, 887-890]. The only reference, which is consistent with the claim that smaller cations demonstrate stronger ion-pairing (Ref. 41), is specifically conducted for proteins, which is quite different from the present case of electrocatalytic CO₂RR, and is not compelling in this context.

Hyungjun Kim

Associate Professor

Department of Chemistry, E6-6 Building, Rm 506
291, Daehak-ro, Yuseong-gu, Daejeon
34141, Republic of Korea

KAIST

Korea Advanced Institute of
Science and Technology

We thank the reviewer to make this keen comment. We first note that our previous discussion pertained to “ion pairs”, although the contracting trend reported in the previous work was related to “contact ion pairs” (a subset of ion pairs). However, we concede that our previous hypothetical explanation based on the ion-pairing trend may still be imperfect; we have thus removed this proposed rationale in a manner consistent with the reviewer’s valid comments.

2. The mechanism of the cation effect (line 209-211), claiming that it originates predominantly from interfacial cation concentration rather than chemical interactions between cations and intermediates, is not sufficiently supported. Essentially, I disagree with the underlying assumption that chemical interactions in the EDL, which are responsible for CO₂RR activation, are not strongly influenced by cation composition.

For example, the authors do not show the calculated electrode potential or dipole moment for Rb⁺, which is actually very important since recent studies have shown Cs⁺ could break the cation-dependent trend in the interfacial solvation structure [JACS Au 2021, 1, 1674–1687], and Rb⁺ represents the turning point. Additionally, the authors should report the coordination numbers and O-M⁺ distance as they did for K⁺, for other cations to show whether these chemical interactions, which influence catalytic performance, are similar between cations (which I believe is not correct). Ideally the authors would also investigate cation concentration dependence in these calculations to show whether these important parameters are significantly impacted by cation density.

We thank the reviewer to point this out. As mentioned in our above response, we note that our finding of local colligativity is not derived from an *assumption* that the interaction strengths of various cationic species are the same, but follows from the experimental *observation* that the catalytic activity is singly correlated with the local cation concentration regardless of the cation identity. This experimental observation dictates that the local concentration is the predominant factor leveling effect of different interaction strengths between different cations, given that such interaction strengths would vary. We argue that all classes of colligative properties, such as a boiling point elevation and osmotic pressure, can be observed for various ionic species, even though the specific interaction between ions and water significantly differs.

Also, as shown in Figures R17a and R17b, the experimental CO₂RR activities for CO and CORR for C₂H₄ generation are singly correlated with the local cation concentration for all of the various cationic species (including the Rb⁺), where the positive correlation is strong enough (but not perfect) to formulate a general trend; a certain extent of scatter can be attributed to the different interaction strengths between ions and water, as can typically be seen from the trend of colligative properties. However, such a positive correlation lacks in the experimental CORR activity for CH₄

production (Figure R17c), which has been identified to be controlled by the PCET, not by the CCET.

We further note that our experimental data included the Rb^+ case, which also obeys the general trend constructed by other cations.

"x" M KOH : ● 0.01, ● 0.1, ● 0.5, ● 1, ● 5, ● 10, ● 5_Nafion
0.01 M KOH + "y" M K_2CO_3 : ⊕ 0.045, ⊗ 0.495,
0.5 M KOH + "y" M K_2CO_3 : ⊙ 0.25, ⊚ 2.25
0.05 M "M" CO_3 : ★ Li, ◆ Na, ⊕ K, ⊙ Rb, ⊚ Cs
1 M "M"OH : ★ Li, ◆ Na, ● K, ⊙ Rb, ⊚ Cs

Figure R17. Reaction kinetic study of CO₂RR and CORR. Correlation plots of a, j_{CO} on the Ag electrode, b, $j_{\text{C}_2\text{H}_4}$ and c, j_{CH_4} on the Cu electrode with $|\sigma|$. The data was collected at $-1.3 V_{\text{SHE}}$ for the CO₂-to-CO formation on Ag electrode, $-1.4 V_{\text{SHE}}$ for the CO-to-C₂H₄ formation, and $-1.65 V_{\text{SHE}}$ for the CO-to-CH₄ formation on the Cu electrode. Electrolytes used for gathering these data can be classified into five different categories: KOH electrolytes with different concentrations (filled circles), 0.01 M KOH electrolytes with additional K₂CO₃ salt (double-crossed circles), 0.5 M KOH electrolytes with additional K₂CO₃ salt (single-crossed circles), 0.05 M M₂CO₃ electrolytes (half-filled symbols), and 1 M MOH electrolytes (filled symbols), where M is Li, Na, K, Rb, and Cs. In addition, a result collected on the Nafion-coated Cu electrode in 5 M KOH electrolyte (a filled circle with pale blue color) was also provided. Here, the total M⁺ concentration and its identity are distinguished by color and symbol shape, respectively. Guidelines for the slopes are indicated by a dashed line. Figures R17a and R17b are included in the “Figure 4” of the revised manuscript.

3. The alkali cations have very different solvation energies, hydrated radii, local solvation structures, polarizabilities, etc. These properties have been extensively investigated, both in the context of electrocatalytic interfaces and in solutions more generally. Here the authors show that CO₂RR activity is first order in cation concentration, leading to the conclusion that the cation effect is a colligative property. However, this conclusion is based on the assumption that specific interactions between cations and solvent as well as between cations and CO₂ is essentially unaffected by cation identity, which is almost certainly not correct. As explained in comment 2 above, this underlying assumption is first not specifically tested, and second contradicts numerous previous reports. This is my overarching concern with the manuscript, and the primary reason I cannot recommend it for publication, despite the interesting correlation between CO₂RR activity and local cation concentration.

We are grateful for the reviewer’s keen point and concerns about other possible phenomena. We believe, however, that our experimental observation of the counter-intuitive phenomenon of emergence of a local colligative property is interesting enough to be published in Nature Communications. Considering the behavior of other well-known colligative properties, such as a boiling-point elevation and an osmotic pressure, which exhibit a general trend dependent only on the concentration regardless of variations in the magnitudes of ion-water interaction strengths, we determine that the short-range electrostatic interactions between the alkali cation and the *CO₂⁻ intermediate are strong enough to level the relative differences among different cation identities. Furthermore, we note that our findings pertaining to local colligative properties can be extended to explain the origin of empirical findings pertaining to the maximization of C₂H₄ production using an ionomer in the context of the CCET-based mechanism (Figure R18).

Figure R18. Boosted C₂H₄ formation on the Nafion-coated Cu electrode. a, The $j_{C_2H_4}$ vs. potential curves and b, the C_{diff} curves, measured on bare and Nafion-coated Cu electrodes in 5 M KOH electrolyte. c, Schematic descriptions of the catalyst–electrolyte interface for the Nafion-free and Nafion-coated Cu electrodes, showing enriched population of M⁺ near the interface on the Nafion-coated one. This is included in the “Figure 5” of the revised manuscript.

4. For the experimental evidence, the authors only perform kinetics measurements for different cations in 0.05 M carbonate solutions. More kinetics measurement in very diluted solutions (where interfacial cation concentration should be similar, not restricted by cation sizes) and very concentrated solutions (where interfacial cation concentration is different) [Energy Environ. Sci., 2019, 12, 3380--3389] should also be conducted for comparison.

We thank the reviewer to point this out. The revised manuscript investigates a wide range of cation concentrations. Notably, the CO₂RR on Ag electrode was conducted using an electrolyte concentration range that varied from 0.01 M to 10 M KOH (Figure R17a).

5. The authors should provide the equivalent circuit they employed to fit the impedance result and explain how they derive the C_{diff} .

We are grateful for the reviewer's comment. We have introduced an RC circuit to fit the impedance results. (Figure R19)

Figure R19. RC circuit for fitting obtained impedance data. The obtained impedance data was fitted by the RC circuit given as $Z = R + 1/i\omega C_{diff}$, where R is the solution resistance, and ω is the circular frequency. It is included in "Supplementary Figure 35" of the revised manuscript.

In summary, although the authors' findings show the importance of cation concentration for CO₂RR via a cation-coupled electron transfer mechanism, the main new conclusion of this work, that the cation effect is a colligative property driven by ion-pairing, is not supported and contradicts many previous experimental and theoretical findings. More careful investigations should be performed, specifically investigating differences in interfacial solvation structure between cations. Due to these concerns regarding both novelty and accuracy of the findings, I do not recommend this work for publication in Nature Communications.

We appreciate the reviewer's thoughtful comments and suggestions, which were exceedingly helpful in revising our manuscript to substantively enhance its impact. In general, our kinetic study is the most direct approach to validating a given reaction mechanism. Our work first offers direct evidence of CCET based on the results of a kinetic study of CO₂RR, while also reporting the interesting experimental observation of a local colligative behavior. Moreover, in the substantially revised manuscript, such findings have been extended from CO formation to other reaction paths, including the formation of CH₄ and C₂H₄, while rationalizing the mechanistic

Complex molecular-Systems
Multiscale Design

D E S I G N

Hyungjun Kim

Associate Professor

Department of Chemistry, E6-6 Building, Rm 506

291, Daehak-ro, Yuseong-gu, Daejeon

34141, Republic of Korea

KAIST

Korea Advanced Institute of
Science and Technology

origin of why an ionomer-existing interface can yield a better activity. In addition to such experimental findings and rationalizations, our revised manuscript also provides important theoretical insights on the nature of CCET, i.e., how it can be similar to or different from the PCET and the unification framework that can be built upon the novel utilization of the concept of “field effect”. We, therefore, believe that our revised manuscript possesses a critical impact on the emerging field of cation effects, warranting its publication in Nature Communications.

REVIEWER COMMENTS

Reviewer #1 (Remarks to the Author):

I congratulate the authors for the significant revision of this work, which now almost entirely fulfills the requirements of novelty and impact. However, before suggesting it for publication, I think additional revisions are needed, some of them related to the scientific content and others to the overall presentation of the work.

MAJOR POINTS

- 1) In my opinion the authors do not give the appropriate credit to the previous work by Monteiro et al. (Nat. Catal. 2021, 4, 654–662) and to Koper's works in general. Certainly Monteiro et al. did not explicitly mention cation-coupled electron transfer (CCET) in their manuscript (although the corresponding author Marc Koper did mention it during his talks), however they introduced the concept. Besides, Monteiro et al. also employed cation-intermediate coordination number as a descriptor of cation effect in both the Nat. Catal. work and a more recent one (J. Am. Chem. Soc. 2022, 144 1589–1602). Shin et al. employed cation-intermediate coordination number repeatedly in this study, but without giving the credit to the original work. As a single example of the systematic lack of credit, see the sentence "Our finding elucidates the origin of empirically-found present optimizations" in the abstract. In my opinion this sentence fully neglects the significant mechanistic advances carried out by Koper et al. in understanding cation effect. As a more general example of recurrent lack of credit, by quickly reading the paper it seems that Shin et al. have been the first to propose cations to stabilize the CO-CO dimer. This is simply not true (see Angew. Chem. Int. Ed., 56, 3621–3624 and ACS Catal. 2021, 11, 12336–12343, which are not even cited).
- 2) I urge the authors to cite the review J. Chem. Phys. 2019, 151, 160902 in the introduction and discuss all the theories about cation effect discussed therein. The current version of the manuscript seems to acknowledge only the mean field and cation coordination theories, which is not correct.
- 3) Could the authors motivate why CCET governs the RDS of CO₂ reduction reaction only from neutral to alkaline pH conditions? Is it due to low cation concentration at the surface under acidic conditions or is there any other reason? Although an unequivocal explanation is beyond the scope, the authors should at least discuss the evidence.
- 4) The authors motivated the use of the ACE scale (which in my opinion should be updated to CCE, see minor points) by referring to the Butler Volmer equation (Supplementary Note 3). As the authors mentioned, that equation involves the concentration of reactant (O) and product (R) during a multi-electron transfer process. As far as I understand the theory, assuming CO₂RR to CO as a case study, O should be CO₂ and R should be CO. Why do the authors directly replace the concentration of species O by the concentration of the cation? Since cation enables CO₂RR, I agree that the concentration of reactive CO₂ (C_O) should be somehow equivalent to surface concentration of cation, however that equivalence should be explicitly mentioned and not taken for granted.

5) I would remove the section of "colligative property", since in my opinion such discussion does not expand the current knowledge on cation effect. It is well-known since the work of Resasco (J. Am. Chem. Soc., 138, 13006–13012, at least) that cation effect for alkali metals is mainly determined by cation concentration, so I do not understand why the authors need to turn something well-understood into a new concept.

6) Did the authors detect ethanol during the CO reduction study on Cu? If so, I would be very interested to see how ethanol partial current density depends on applied potential vs ACE scale. Such study would either confirm or invalidate the current hypothesis that CO-CO dimerization is the RDS toward all the C₂+ products.

7) I disagree with the Data availability statement as not all the data are available in the main text or the SI. I urge instead the authors to upload all the datasets to open source database as ioChem-BD (DFT) and Zenodo (experiments).

MINOR POINTS

1) I suggest the authors to update the ACE notation (Activity-corrected electrode) with CCE (cation concentration-corrected electrode)

2) I suggest the authors to keep the same y-axis in graphs related to cation-intermediate coordination number (as Supplementary Fig. 4 and 5). Currently, it is very hard to compare different cases as the scale continuously change.

3) In Supplementary Figure 10 the ethylene is represented as a oxalate (carbon + oxygen) instead of as C₂H₄.

Reviewer #3 (Remarks to the Author):

While I appreciate the significant revisions performed by the authors, I still have a remaining concern that prevents me from recommending this article for publication in its current form. However, if this can be appropriately addressed, I believe this paper could be reconsidered for publication in Nature Communications.

It is clear from experimental data that interfacial solvation plays a critical role in reaction kinetics. Interfacial solvation is strongly cation dependent. [<https://doi.org/10.1063/1.5124878>; <https://doi.org/10.1038/s41929-022-00816-0>] Interfacial electric fields also depend on cation identity and the associated interfacial solvation structure. [<https://doi.org/10.1021/acs.jpcc.2c01134>; <https://doi.org/10.1021/jacsau.1c00512>]. The kinetic measurements by these authors reproduce numerous other studies showing that for a fixed bulk concentration, reaction kinetics for CO₂ reduction on Ag and Cu are cation dependent (see Figures S24 and S28). From this study, it is not possible to rule

out that any of the above effects contribute to this observation, which do not necessarily represent colligative properties.

First, in this study, there is no experimental measurement or direct theoretical prediction for how the local cation concentration at the interface varies as a function of cation. The authors themselves even state: "Unfortunately, this parameter is not straightforwardly measurable or even defined."

Second, the theory performed in this study considers only a single unsolvated cation with no interfacial water. No effort is made to vary the identity of the cation or its effect on interfacial solvation. Consequently, this work does not evaluate any of the probable chemical origins for specific cation effects.

Third, the authors rely on Butler-Volmer kinetics to substantiate their hypothesis that cation influence on reaction kinetics is a colligative property. However, Butler-Volmer kinetics consider only the surface charge density and do not account for specific cation effects, so this is a circular argument.

Fourth, it is known that cation-specific interfacial solvation, like reaction kinetics is strong surface site dependent. [<https://doi.org/10.1039/D2SC01878K>] This also is not accounted for in Butler-Volmer kinetics even though it has an order of magnitude or greater influence on the site-specific reaction rate.

Consequently, the conclusion that cation effects represent a colligative property is not supported by the current work and also contradicts numerous previous studies. Even in the absence of this claim, the current theory provides a unifying framework for understanding cation concentration effects in CO₂ and CO reduction, so I am not sure why the authors are unwilling to modify this assertion. The change in title is appropriate; however, I still cannot recommend this paper for publication given the remaining section concluding that cation effects represent a colligative property. Before this paper should be accepted, the authors need to make a substantive effort to summarize competing hypotheses for specific cation effects, including those listed above. They should also acknowledge that without directly characterizing the local cation concentration at the interface, it cannot be conclusively established whether cation effects in CO₂R are a colligative property or not.

Hyungjun Kim

Associate Professor

Department of Chemistry, E6-6 Building, Rm 506
291, Daehak-ro, Yuseong-gu, Daejeon
34141, Republic of Korea

KAIST

Korea Advanced Institute of
Science and Technology

Point-by-point responses to the reviewers

Comments to Reviewer. Thank you for your helpful comments, which are reproduced here in *italics*. Our responses are in **boldface**.

Reviewer #1

I congratulate the authors for the significant revision of this work, which now almost entirely fulfills the requirements of novelty and impact. However, before suggesting it for publication, I think additional revisions are needed, some of them related to the scientific content and others to the overall presentation of the work.

We would like to thank the reviewer for the favorable comments and recognition of the significance of our work. The additional points were revised following the reviewer's guidance.

1) *In my opinion the authors do not give the appropriate credit to the previous work by Monteiro et al. (Nat. Catal. 2021, 4, 654–662) and to Koper's works in general. Certainly Monteiro et al. did not explicitly mention cation-coupled electron transfer (CCET) in their manuscript (although the corresponding author Marc Koper did mention it during his talks), however they introduced the concept. Besides, Monteiro et al. also employed cation-intermediate coordination number as a descriptor of cation effect in both the Nat. Catal. work and a more recent one (J. Am. Chem. Soc. 2022, 144 1589–1602). Shin et al. employed cation-intermediate coordination number repeatedly in this study, but without giving the credit to the original work. As a single example of the systematic lack of credit, see the sentence "Our finding elucidates the origin of empirically-found present optimizations" in the abstract. In my opinion this sentence fully neglects the significant mechanistic advances carried out by Koper et al. in understanding cation effect. As a more general example of recurrent lack of credit, by quickly reading the paper it seems that Shin et al. have been the first to propose cations to stabilize the CO-CO dimer. This is simply not true (see Angew. Chem. Int. Ed., 56, 3621–3624 and ACS Catal. 2021, 11, 12336–12343, which are not even cited).*

We apologize for our lack of sensitivity. However, the phrase “empirically-found present optimization” meant the use of a Nafion-coated electrode, which has not been explained yet to the best of our knowledge. To avoid any confusion, we revised the sentence to

Hyungjun Kim

Associate Professor

Department of Chemistry, E6-6 Building, Rm 506
291, Daehak-ro, Yuseong-gu, Daejeon
34141, Republic of Korea

KAIST

Korea Advanced Institute of
Science and Technology

“Our finding further rationalizes the merit of using Nafion-coated electrode for enhanced C2 production in terms of enhanced surface charge density.”

Furthermore, to properly cite and highlight previous seminal works, we added the following sentence to the “Nature of cation-coupled electron transfer” section;

“The results of DFT-CES simulation are in agreement with the previous novel findings showing that the alkali cations can stabilize the *CO₂ intermediate [*Nat. Catal.* 2021, 4, 654–662 and *J. Am. Chem. Soc.* 2022, 144, 1589–1602] and *OCCO intermediate [*Angew. Chem. Int. Ed.* 2017, 56, 3621–3624 and *ACS Catal.* 2021, 11, 12336–12343].”

2) *I urge the authors to cite the review J. Chem. Phys. 2019, 151, 160902 in the introduction and discuss all the theories about cation effect discussed therein. The current version of the manuscript seems to acknowledge only the mean field and cation coordination theories, which is not correct.*

We thank the reviewer for his or her thoughtful suggestion. In the revised manuscript, we tried to cover many existing theories about the cation effect based on the review paper provided by the reviewer. Exemplifying, we added the below part to the Introduction section:

“Moreover, cation-dependent interfacial water structure has been exploited to understand the cation effect on the CO₂RR, which yields different electric field strengths [*JACS Au* 2022, 2, 472–482 and *J. Phys. Chem. C* 2022, 126, 8477–8488], adsorption rate [*Nat. Catal.* 2022, 5, 624–632], or surface-dependent solvation structure [*Chem. Sci.* 2022, 13, 7634–7643]. There further exist other general discussions on the cation effect to the electrocatalytic activity [*J. Chem. Phys.* 2019, 151, 160902], *e.g.*, site-blocking of reactants on the electrode or surface reconstruction, albeit it has not been directly linked with the cation effect on the CO₂RR.”

3) *Could the authors motivate why CCET governs the RDS of CO₂ reduction reaction only from neutral to alkaline pH conditions? Is it due to low cation concentration at the surface under acidic conditions or is there any other reason? Although an unequivocal explanation is beyond the scope, the authors should at least discuss the evidence.*

In acidic conditions, the alkali metal cation is not a unitary unit compensating for the negatively charged electrode surface, but the proton (or hydronium ion) competes with the alkali metal cation [*J. Am. Chem. Soc.* 2005, 127, 15916–15922]. Thus, we expect a retarded accumulation of

Hyungjun Kim

Associate Professor

Department of Chemistry, E6-6 Building, Rm 506
291, Daehak-ro, Yuseong-gu, Daejeon
34141, Republic of Korea

KAIST

Korea Advanced Institute of
Science and Technology

the alkali metal cation at the vicinity of electrode surface, yielding a lack of CCET mechanism in the acidic condition. We added this discussion to the section of “Cation concentration-dependent Nernstian shifts”;

“However, the CO₂RR in acidic electrolytes is unable to be explained together due to a partial displacement of cations by the protons (or hydronium ions) [*J. Am. Chem. Soc.* 2005, 127, 15916–15922]”.

4) *The authors motivated the use of the ACE scale (which in my opinion should be updated to CCE, see minor points) by referring to the Butler Volmer equation (Supplementary Note 3). As the authors mentioned, that equation involves the concentration of reactant (O) and product (R) during a multi-electron transfer process. As far as I understand the theory, assuming CO₂RR to CO as a case study, O should be CO₂ and R should be CO. Why do the authors directly replace the concentration of species O by the concentration of the cation? Since cation enables CO₂RR, I agree that the concentration of reactive CO₂ (C_O) should be somehow equivalent to surface concentration of cation, however that equivalence should be explicitly mentioned and not taken for granted.*

Following the reviewer’s guidance, we included more details in the Supplementary Note 3;

“For the case of CO₂RR or CORR, there are more than one molecule at the reactant side of the RDS (see (R1) and (R3) of the manuscript), and thus the C_O(0) term of (Eq. S14) needs to be modified into C_{CO₂}(0)C_{M⁺}(0) or C_{CO}²(0)C_{M⁺}(0), respectively. Here C_{CO₂}(0), C_{CO}(0), and C_{M⁺}(0) denote the local concentration of CO₂, CO, and M⁺ at the electrode surface, respectively. Considering a local equilibrium between the gaseous reactant and its local concentration, then, the C_{CO₂}(0) and C_{CO}(0) can be written as K₁P_{CO₂} and K₂P_{CO}, respectively (where K₁ and K₂ are corresponding equilibrium constants).”

5) *I would remove the section of "colligative property", since in my opinion such discussion does not expand the current knowledge on cation effect. It is well-known since the work of Resasco (*J. Am. Chem. Soc.*, 138, 13006–13012, at least) that cation effect for alkali metals is mainly determined by cation concentration, so I do not understand why the authors need to turn something well-understood into a new concept.*

Hyungjun Kim

Associate Professor

Department of Chemistry, E6-6 Building, Rm 506
291, Daehak-ro, Yuseong-gu, Daejeon
34141, Republic of Korea

KAIST

Korea Advanced Institute of
Science and Technology

By accepting the reviewer's reasonable point, the discussion related to the local colligative property and the consequent interpretation of CO₂RR activity are now removed. In the revised manuscript, the discussion is based on the activity correlation with the surface charge density, which is considered to be the new addition to the previous understanding.

6) *Did the authors detect ethanol during the CO reduction study on Cu? If so, I would be very interested to see how ethanol partial current density depends on applied potential vs ACE scale. Such study would either confirm or invalidate the current hypothesis that CO-CO dimerization is the RDS toward all the C₂+ products.*

We are grateful for the keen suggestion. We are also well aware of the importance of progressive investigations from the current CO, CH₄, and C₂H₄ productions to more complex but valuable products such as methanol, ethanol, etc. Unfortunately, ethanol is not a major product on our polycrystalline Cu electrode. Thus, we could not clearly compare their polarization curves upon the CCE (or, in the original version, ACE) scale. Since the production of ethanol typically requires fabrications of more advanced electrode materials such as the oxide-driven Cu electrode [*Nat. Commun.* 2022, 13, 3754 and *Nature* 2014, 508, 504–507], we have collaborated with Prof. Dae Hyun Nam's group at DGIST (one of the authors in this work), who has a strong background in synthesizing the highly active electrode materials toward the CO₂RR. This is an ongoing project, and we hope to share the results of this project in the near future.

7) I disagree with the Data availability statement as not all the data are available in the main text or the SI. I urge instead the authors to upload all the datasets to open source database as ioChem-BD (DFT) and Zenodo (experiments).

The statement at the Data availability is revised as “All data is available in the main text or Supplementary Information. The simulation results are uploaded to the ioChem-BD and experimental results are uploaded to the Zenodo.” All the data will be released when the revision process is finished and the statement will be accordingly revised in the final version of manuscript.

Hyungjun Kim

Associate Professor

Department of Chemistry, E6-6 Building, Rm 506

291, Daehak-ro, Yuseong-gu, Daejeon

34141, Republic of Korea

KAIST

Korea Advanced Institute of
Science and Technology

Minor 1) I suggest the authors to update the ACE notation (Activity-corrected electrode) with CCE (cation concentration-corrected electrode)

We agree to the reviewer, and thus we modified the ACE notation to the CCE in the revised manuscript.

Minor 2) I suggest the authors to keep the same y-axis in graphs related to cation-intermediate coordination number (as Supplementary Fig. 4 and 5). Currently, it is very hard to compare different cases as the scale continuously change.

We accepted the reviewer's suggestion and now the coordination number can be clearly compared with different intermediate species in Figure 1 and Supplementary Fig. 3-5.

Minor 3) In Supplementary Figure 10 the ethylene is represented as a oxalate (carbon + oxygen) instead of as C_2H_4 .

We apologize for our oversight, and revised the Supplementary Figure 10.

Hyungjun Kim

Associate Professor

Department of Chemistry, E6-6 Building, Rm 506
291, Daehak-ro, Yuseong-gu, Daejeon
34141, Republic of Korea

KAIST

Korea Advanced Institute of
Science and Technology

Reviewer #3

While I appreciate the significant revisions performed by the authors, I still have a remaining concern that prevents me from recommending this article for publication in its current form. However, if this can be appropriately addressed, I believe this paper could be reconsidered for publication in Nature Communications.

We would like to thank the reviewer for the favorable comments and recognition of the significance of our work. We also thank for his or her sincere comments to further improve our manuscripts, which are now properly reflected to the revised manuscript.

It is clear from experimental data that interfacial solvation plays a critical role in reaction kinetics. Interfacial solvation is strongly cation dependent. [<https://doi.org/10.1063/1.5124878>; <https://doi.org/10.1038/s41929-022-00816-0>] Interfacial electric fields also depend on cation identity and the associated interfacial solvation structure. [<https://doi.org/10.1021/acs.jpcc.2c01134>; <https://doi.org/10.1021/jacsau.1c00512>]. The kinetic measurements by these authors reproduce numerous other studies showing that for a fixed bulk concentration, reaction kinetics for CO₂ reduction on Ag and Cu are cation dependent (see Figures S24 and S28). From this study, it is not possible to rule out that any of the above effects contribute to this observation, which do not necessarily represent colligative properties.

We thank the reviewer for his or her thoughtful suggestion. In the revised manuscript, we tried to cover many existing theories about the cation effect based on the papers provided by the reviewer. Exemplifying, we added the below part to the Introduction section:

“Moreover, cation-dependent interfacial water structure has been exploited to understand the cation effect on the CO₂RR, which yields different electric field strengths [*JACS Au* 2022, 2, 472–482 and *J. Phys. Chem. C* 2022, 126, 8477–8488], adsorption rate [*Nat. Catal.* 2022, 5, 624–632], or surface-dependent solvation structure [*Chem. Sci.* 2022, 13, 7634–7643]. There further exist other general discussions on the cation effect to the electrocatalytic activity [*J. Chem. Phys.* 2019, 151, 160902], *e.g.*, site-blocking of reactants on the electrode or surface reconstruction, albeit it has not been directly linked with the cation effect on the CO₂RR.”

Hyungjun Kim

Associate Professor

Department of Chemistry, E6-6 Building, Rm 506
291, Daehak-ro, Yuseong-gu, Daejeon
34141, Republic of Korea

KAIST

Korea Advanced Institute of
Science and Technology

First, in this study, there is no experimental measurement or direct theoretical prediction for how the local cation concentration at the interface varies as a function of cation. The authors themselves even state: “Unfortunately, this parameter is not straightforwardly measurable or even defined.”

We agree with the reviewer’s keen point. We thus removed the discussion related to the local colligative property and the consequent interpretation of CO₂RR activity in the revised manuscript.

Second, the theory performed in this study considers only a single unsolvated cation with no interfacial water. No effort is made to vary the identity of the cation or its effect on interfacial solvation. Consequently, this work does not evaluate any of the probable chemical origins for specific cation effects.

In continuation of our response, we mostly agree with the reviewer’s criticism, and thus removed the discussion related to the local colligative property. However, we would like to point out that our DFT-CES simulation explicitly considers full atomic details of the electrolyte phase, and thus include the solvated water structure around the cations. We suspect that some confusion might originate from the Figures 1 and 2, where the water molecules were not explicitly shown for visual clarity. To avoid any further confusion, we clearly indicated the omittance of water molecules in the revised captions of Figures 1 and 2.

Third, the authors rely on Butler-Volmer kinetics to substantiate their hypothesis that cation influence on reaction kinetics is a colligative property. However, Butler-Volmer kinetics consider only the surface charge density and do not account for specific cation effects, so this is a circular argument.

To reflect the reviewer’s concern, we revised our manuscript to be based on the activity correlation with the surface charge density (instead of the local concentration), which is what actually measured from our experiments.

Fourth, it is known that cation-specific interfacial solvation, like reaction kinetics is strong surface site dependent. [<https://doi.org/10.1039/D2SC01878K>] This also is not accounted for in Butler-Volmer kinetics even though it has an order of magnitude or greater influence on the site-specific reaction rate.

Hyungjun Kim

Associate Professor

Department of Chemistry, E6-6 Building, Rm 506
291, Daehak-ro, Yuseong-gu, Daejeon
34141, Republic of Korea

KAIST

Korea Advanced Institute of
Science and Technology

As mentioned above, we discussed about the importance of cation-dependent interfacial solvation structure by adding the following sentences;

“Moreover, cation-dependent interfacial water structure has been exploited to understand the cation effect on the CO₂RR, which yields different electric field strengths [*JACS Au* 2022, 2, 472–482 and *J. Phys. Chem. C* 2022, 126, 8477–8488], adsorption rate [*Nat. Catal.* 2022, 5, 624–632], or surface-dependent solvation structure [*Chem. Sci.* 2022, 13, 7634–7643]. There further exist other general discussions on the cation effect to the electrocatalytic activity [*J. Chem. Phys.* 2019, 151, 160902], *e.g.*, site-blocking of reactants on the electrode or surface reconstruction, albeit it has not been directly linked with the cation effect on the CO₂RR.”

To suggest a valuable research direction for further studies, we also provided some discussion about a potential link between the cation-dependent surface charge density and the cation-specific interfacial solvation structure;

“This enables us to conclude that the cation effect on CO and C₂H₄ formations to be controlled by the $|\sigma|$. However, the atomic origin of its cation-species dependence needs to be further unraveled; cation-specific interfacial solvation could play a significant role [*JACS Au* 2022, 2, 472–482, *J. Phys. Chem. C* 2022, 126, 8477–8488, *Nat. Catal.* 2022, 5, 624–632, *Chem. Sci.*, 2022, 13, 7634–7643, *J. Chem. Phys.* 2019, 151, 160902, *Nat. Catal.* 2021, 4, 654–662, *J. Am. Chem. Soc.* 2022, 144, 1589–1602], suggesting a future research direction.”

Consequently, the conclusion that cation effects represent a colligative property is not supported by the current work and also contradicts numerous previous studies. Even in the absence of this claim, the current theory provides a unifying framework for understanding cation concentration effects in CO₂ and CO reduction, so I am not sure why the authors are unwilling to modify this assertion. The change in title is appropriate; however, I still cannot recommend this paper for publication given the remaining section concluding that cation effects represent a colligative property. Before this paper should be accepted, the authors need to make a substantive effort to summarize competing hypotheses for specific cation effects, including those listed above. They should also acknowledge that without directly characterizing the local cation concentration at the interface, it cannot be conclusively established whether cation effects in CO₂R are a colligative property or not.

We agree that the local concentration is not a directly measured quantity from our experiment, and thus fully admit the reviewer’s keen criticism about the local colligative property. Accordingly, we revised our manuscript to discuss our experimental finding of the kinetic

Hyungjun Kim

Associate Professor

Department of Chemistry, E6-6 Building, Rm 506
291, Daehak-ro, Yuseong-gu, Daejeon
34141, Republic of Korea

KAIST

Korea Advanced Institute of
Science and Technology

dependence on the directly measurable quantity of the surface charge density $|\sigma|$, without trying to draw any hasty claim. We also summarized existing various theories for the specific cation effects (e.g., acidity, field-effect, interfacial solvation structure, site-blocking, surface reconstruction and chemical interaction) in the revised manuscript. Thanks to the reviewer's invaluable suggestion, we believe that our revised manuscript has a more balanced viewpoint, which is now believed to be suitable for publication.